# Categorical Reparameterization with Denoising Diffusion Models

**Samson Gourevitch** [1]   **Alain Durmus** [1]   **Eric Moulines** [2]   **Jimmy Olsson** [3]   **Yazid Janati** [4]

## Abstract

Learning models with categorical variables requires optimizing expectations over discrete distributions, a setting in which stochastic gradient-based optimization is challenging due to the non-differentiability of categorical sampling. A common workaround is to replace the discrete distribution with a continuous relaxation, yielding a smooth surrogate that admits reparameterized gradient estimates via the reparameterization trick. Building on this idea, we introduce REDGE, a novel and efficient diffusion-based soft reparameterization method for categorical distributions. Our approach defines a flexible class of gradient estimators that includes the STRAIGHT-THROUGH estimator as a special case. Experiments spanning latent variable models and inference-time reward guidance in discrete diffusion models demonstrate that REDGE consistently matches or outperforms existing gradient-based methods. Code is available at https://github.com/samsongourevitch/redge.

## 1. Introduction

Many learning problems involve *discrete* choices, such as actions in reinforcement learning, categorical latent variables in variational inference, token-level decisions in sequence modeling, or combinatorial assignments in structured prediction and discrete optimization. A common primitive in these settings is the minimization of an objective of the form $\mathbb{E}_{\pi_\theta}[f(X)]$, where $\pi_\theta$ is a categorical distribution corresponding to the law of $L$ independent discrete random variables each taking values in a vocabulary of size $K$. The function $f$ represents a downstream loss or constraint penalty

[1]CMAP, Ecole polytechnique [2]Mohamed Bin Zayed University of AI and LRE, EPITA [3]KTH Royal Institute of Technology [4]Institute of Foundation Models, MBZUAI. Correspondence to: Samson Gourevitch <samson.gourevitch@polytechnique.edu>, Yazid Janati <yazid.janati@mbzuai.ac.ae>.

*Proceedings of the 43rd International Conference on Machine Learning*, Seoul, South Korea. PMLR 306, 2026. Copyright 2026 by the author(s).

evaluated on discrete samples, typically through one-hot encodings. Computing $\nabla_\theta \mathbb{E}_{\pi_\theta}[f(X)]$ exactly is generally intractable: in the absence of exploitable structure in $f$, it requires summing over $K^L$ configurations. The challenge, therefore, is to construct gradient estimators that are both computationally feasible and have a low mean squared error.

Existing estimators exhibit a standard bias–variance trade-off. Score-function estimators, such as REINFORCE (Williams, 1992; Greensmith et al., 2004), are unbiased but often suffer from high variance, which motivates the use of variance-reduction techniques most often using learned control variates (Tucker et al., 2017; Grathwohl et al., 2018); they often yield useful gradients in practice but are biased with respect to the true discrete objective, with recent refinements such as REINMAX improving the approximation (Liu et al., 2023a). Continuous relaxations based on approximate reparameterizations, most notably the GUMBEL-SOFTMAX / Concrete construction (Maddison et al., 2017; Jang et al., 2017), replace $\pi_\theta$ by a smooth family on the simplex controlled by a temperature parameter. While this enables pathwise differentiation, taking the temperature small to reduce bias drives the sampler towards an argmax map and typically leads to ill-conditioned or vanishing gradients, while higher temperatures provide well-behaved gradients but correspond to optimizing a more relaxed objective.

In this work, we revisit continuous relaxations through the lens of denoising diffusion models (Sohl-Dickstein et al., 2015; Song & Ermon, 2019; Ho et al., 2020). Diffusion models generate data by transforming a Gaussian sample into a sample from the target data distribution through iterative denoising dynamics explicitly constructed as the reverse of a chosen forward noising process. In practice, implementing the sampler requires only access to a denoiser; that is, a function that, given a noisy input and its noise level or time index, returns the expected clean signal.

**Contributions.** We exploit the key observation that for a categorical distribution supported on simplex vertices, the denoiser at each noise level can be computed in closed form. This enables us to construct a training-free, diffusion-based, differentiable, and approximate sampling map from Gaussian noise to the categorical distribution $\pi_\theta$. We then analyze the small-noise regime, which serves as the temperature parameter, and characterize the emergence of nearly

constant transport regions and sharp decision boundaries, explaining when and why gradients become uninformative as the relaxation approaches the discrete target. We derive practical gradient estimators, including hard variants, that recover the hard STRAIGHT-THROUGH (Bengio et al., 2013) and REINMAX (Liu et al., 2023a) as special cases when using a single diffusion step. We also propose a parameter-dependent initialization that improves performance while keeping the diffusion overhead small. Empirically, we find that our diffusion-based reparameterization yields strong results across a diverse set of benchmarks, including polynomial programming, variational inference, and inference-time reward guidance, typically matching or improving upon prior estimators.

**Notation.** For any positive integer $n$, let $[n] := \{1, \ldots, n\}$. We denote the $K$-simplex by $\Delta^{K-1} = \{w \in \mathbb{R}_+^K : \sum_{k=1}^{K} w^k = 1\}$. For a matrix $x \in \mathbb{R}^{L \times K}$, we write $x^i \in \mathbb{R}^K$ for its $i$-th row and $x^{ij}$ for the $(i, j)$-th element. Besides, we identify any $x \in \mathbb{R}^{L \times K}$ as a vector using the row-major order in which the matrix elements are ordered by row. The softmax operator on a matrix $x \in \mathbb{R}^{L \times K}$ is defined row-wise by $\text{softmax}(x) \in \mathbb{R}^{L \times K}$ with entries $\text{softmax}(x)^{ik} = \exp(x^{ik})/\sum_{j=1}^{K} \exp(x^{ij})$ for $(i, k) \in [L] \times [K]$. For a map $f : \mathbb{R}^d \to \mathbb{R}^m$, we write $J_x f \in \mathbb{R}^{m \times d}$ for its Jacobian matrix. To write Jacobians for maps $f : \mathbb{R}^{L' \times K'} \to \mathbb{R}^{L \times K}$ conveniently, we implicitly identify matrices with their vectorized forms. Gradients and Jacobians are taken with respect to these vectorized representations, and we do not distinguish notationally between a matrix and its vectorization.

## 2. Background

We consider optimization problems where the objective is an expectation with respect to a *discrete* distribution over a finite vocabulary $\mathsf{X}$, of the form

$$F(\theta) = \mathbb{E}_{\pi_\theta}[f_\theta(X)] := \sum_{x \in \mathsf{X}} f_\theta(x)\, \pi_\theta(x)\,, \qquad (1)$$

where $f : \mathsf{X} \times \Theta \to \mathbb{R}$, $\Theta \subseteq \mathbb{R}^m$, and $\{\pi_\theta : \theta \in \Theta\}$ is a parameterized family of probability mass functions (p.m.f.) over $\mathsf{X}$. Without loss of generality, we assume that $\mathsf{X} = \mathsf{V}^L$ for some $L \in \mathbb{N}$, where $\mathsf{V} := \{e_k\}_{k=1}^{K}$ denotes the set of $K$ one-hot encodings, and $e_k \in \mathbb{R}^K$ is the one-hot vector with 1 at position $k$. We also assume that the distribution $\pi_\theta$ factorizes according to this categorical structure: for any $\theta \in \Theta$ and $x = (x^1, \ldots, x^L) \in \mathsf{X}$,

$$\pi_\theta(x) = \prod_{i=1}^{L} \pi_\theta^i(x^i),\ \pi_\theta^i(x^i) := \frac{\exp(\langle x^i, \varphi_\theta^i \rangle)}{\sum_{j=1}^{K} \exp(\varphi_\theta^{ij})}\,, \quad (2)$$

where $\theta \mapsto \varphi_\theta \in \mathbb{R}^{L \times K}$ is such that $\varphi_\theta^i$ are the logits of the $i$-th categorical component. The factorization (2) is standard

and is used in reinforcement learning to model policies (Wu et al., 2018; Berner et al., 2019; Vinyals et al., 2019), in training Boltzmann machines (Hinton, 2012), in VQ-VAEs (Van Den Oord et al., 2017), and more recently for modelling transitions in discrete diffusion models.(Hoogeboom et al., 2021; Austin et al., 2021; Campbell et al., 2022; Lou et al., 2023; Shi et al., 2024; Sahoo et al., 2024).

Under mild regularity assumptions on $f$ and $\varphi_\theta$, the gradient of (1) is given by

$$\nabla_\theta F(\theta) = \mathbb{E}_{\pi_\theta}[\nabla_\theta f_\theta(X)] + \sum_{x \in \mathsf{X}} f_\theta(x)\nabla_\theta \pi_\theta(x) \quad (3)$$

and is intractable as the sum ranges over $K^L$ states. Furthermore, while the first term can be approximated via Monte Carlo, the second term has to be estimated separately. One option is to use the REINFORCE estimator. (Williams, 1992) However, it is well known that the vanilla forma of this estimator suffers from high variance (Sutton & Barto, 2018) and has be to combined with baselines or other control-variate techniques to reduce variance (Greensmith et al., 2004; Mnih & Gregor, 2014; Mnih & Rezende; Tucker et al., 2017; Titsias & Shi, 2022; Grathwohl et al., 2018). Other estimation methods have been proposed, such as the STRAIGHT-THROUGH estimator (Bengio et al., 2013) or GUMBEL-SOFTMAX reparameterization (Maddison et al., 2017; Jang et al., 2017), which we briefly review here.

For simplicity, we assume throughout that $f$ does not depend on $\theta$ (i.e., $f_\theta(x) = f(x)$) and is differentiable in $x$.

**STRAIGHT-THROUGH and REINMAX estimators.** Popular estimators either replace the objective $F$ by a differentiable surrogate and use its gradient, or directly construct a surrogate for $\nabla_\theta F$ itself. One such estimator is the STRAIGHT-THROUGH (ST) approach, which replaces the discrete objective by the surrogate obtained by swapping $f$ and the expectation in (1), and differentiates the map $\theta \mapsto f(\mathbb{E}_{\pi_\theta}[X])$. Noting that $J_\theta \mathbb{E}_{\pi_\theta}[X] = \mathbb{C}\text{ov}_{\pi_\theta}(X) J_\theta \varphi_\theta$, the gradient of this surrogate is $J_\theta \varphi_\theta^\top \mathbb{C}\text{ov}_{\pi_\theta}(X)\nabla_x f(\mathbb{E}_{\pi_\theta}[X])$. A popular practical instance of ST replaces the expectation inside $\nabla_x f$ with a single Monte Carlo sample $X \sim \pi_\theta$, often referred to as *hard* ST:

$$\widehat{\nabla}_\theta^{\text{ST}} F(X; \theta) := J_\theta \varphi_\theta^\top \mathbb{C}\text{ov}_{\pi_\theta}(X)\nabla_x f(X)\,. \quad (4)$$

This gradient estimator was first considered by Hinton et al. (2012) in the context of training with hard thresholds, where the backward pass treats the threshold operation as the identity. It was later formalized by Bengio et al. (2013) for quantization-aware training of deep networks. The resulting gradient estimator is often effective in practice but is, by construction, biased with respect to the true discrete objective. When $f$ is linear, hard ST yields an unbiased gradient of $F$.

The REINMAX estimator (Liu et al., 2023a) refines hard ST by providing an exact unbiased estimator of $\nabla_\theta F(\theta)$ in case $f$ is quadratic, obtained via a trapezoidal (Heun-type) rule. In Appendix B, we show that it admits the following simple form, closely mirroring hard ST:

$$\widehat{\nabla}_\theta^{\mathrm{RM}} F(X; \theta) := \frac{1}{2} \, \mathrm{J}_\theta \varphi_\theta^\top B_\theta(X) \nabla_x f(X) \, , \qquad (5)$$

where $X \sim \pi_\theta$. Here $B_\theta(X) = \mathbb{C}\mathrm{ov}_{\pi_\theta}(X) + \widehat{C}_\theta(X)$, where $\widehat{C}_\theta(X)$ is block-diagonal with $L$ blocks of size $K \times K$; its $\ell$-th block is $\widehat{C}_\theta^{(\ell)}(X) := (X^\ell - \mathbb{E}_{\pi_\theta}[X^\ell])(X^\ell - \mathbb{E}_{\pi_\theta}[X^\ell])^\top$, implying that $\mathbb{E}_{\pi_\theta}[\widehat{C}_\theta^{(\ell)}(X)] = \mathbb{C}\mathrm{ov}_{\pi_\theta}(X^\ell)$. We provide further details and proofs in Appendix B.

**Continuous relaxations and soft reparameterizations.** For continuous distributions, the reparameterization trick expresses a sample as a deterministic transform of auxiliary noise (Kingma & Welling, 2013). Specifically, we temporarily assume that $\pi_\theta$ is a distribution that admits a reparameterization, that is, $\pi_\theta := \mathrm{Law}(T_\theta(Z))$, where $Z$ follows a distribution $p$ that does not depend on $\theta$, typically uniform or Gaussian. Assume also that for $p$-almost every $z$ the map $\theta \mapsto T_\theta(z)$ is differentiable for any $\theta \in \Theta$ and that $z \mapsto \nabla_\theta f(T_\theta(z))$ satisfies standard domination conditions for all $\theta \in \Theta$, so that differentiation under the expectation is justified by the Lebesgue dominated convergence theorem. Then

$$\begin{aligned} \nabla_\theta F(\theta) &= \nabla_\theta \mathbb{E}[f(T_\theta(Z))] \\ &= \mathbb{E}[\mathrm{J}_\theta T_\theta(Z)^\top \nabla_x f(T_\theta(Z))] \, , \end{aligned} \qquad (6)$$

which yields a low-variance Monte Carlo estimator of the objective gradient (Schulman et al., 2015). In the discrete case however, such an exact reparameterization is not available. Any representation of $\pi_\theta$ as the pushforward of a simple continuous base distribution typically yields a map $\theta \mapsto T_\theta(z)$ that is piecewise constant with jump discontinuities. As a consequence, for any $\theta$, $\mathrm{J}_\theta T_\theta(z) = 0$ for almost every $z$, and therefore $\mathbb{E}[\mathrm{J}_\theta T_\theta(Z)^\top \nabla_x f(T_\theta(Z))] = 0$ while $\nabla_\theta F(\theta) \neq 0$. Thus (6) does not hold in the discrete setting as the differentiability at *every* $\theta \in \Theta$ fails, and the domination condition needed to justify differentiation under the integral sign is violated. To circumvent this issue, one typically resorts to continuous relaxations of $\pi_\theta$, *i.e.*, distributions admitting a density with respect to the Lebesgue measure. For such relaxations, (6) is valid, at the cost of introducing bias in exchange for lower-variance gradient estimates.

The Gumbel–Softmax (or Concrete) distribution (Maddison et al., 2017; Jang et al., 2017) is a canonical example of such a relaxation: $\pi_\theta$ is replaced with a temperature-indexed family of continuous distributions $(\pi_\tau^\theta)_{\tau > 0}$ on the simplex

that admit pathwise gradient estimator satisfying (6). Specifically, $\pi_\tau^\theta := \mathrm{Law}(T_\tau^\theta(G))$ is used as a relaxed surrogate for $\pi_\theta$, where for all $\theta \in \Theta$,

$$T_\tau^\theta(G) := \mathrm{softmax}\big((\varphi_\theta + G)/\tau\big) \, , \quad \tau > 0 \, ,$$

and $G \in \mathbb{R}^{L \times K}$ is a random matrix with i.i.d. Gumbel entries $G^{ij} \sim \mathrm{Gumbel}(0, 1)$. As $\tau \to 0$, $\{\pi_\tau^\theta \, : \, \tau > 0\}$ converges in distribution to $\pi_\theta$; see Gumbel (1954). This is known as the Gumbel-max trick (Maddison et al., 2017). It is easy to verify that for the surrogate objective $F_\tau(\theta) = \mathbb{E}_{\pi_\tau^\theta}[f(X)]$, which converges to $F$ as $\tau \to 0$, (6) holds under appropriate assumptions on $f$, thus allowing an approximate reparameterization trick at the expense of a certain bias controlled by the parameter $\tau$.

**Remark 1.** *To compute the true gradient $\nabla_\theta F(\theta)$, only the values of $f$ on $\mathsf{X}$ are relevant. In contrast, the estimators considered here differentiate a fixed continuous extension of $f$ to $(\Delta^{K-1})^L$ (or to $\mathbb{R}^{L \times K}$) which we assume is provided by the downstream model and denote again by $f$. Note however that distinct extensions may agree on $\mathsf{X}$ while inducing different gradients on the simplex. Throughout, we avoid this ambiguity by treating the choice of extension as part of the problem specification.*

## 3. Method

In this section, we present REDGE (Reparameterized Diffusion Gradient Estimator), which builds on diffusion models to define a continuous relaxation for $\pi_\theta$. We begin by reviewing the basics of these models.

### 3.1. Diffusion models.

We present denoising diffusion models (DDMs) (Sohl-Dickstein et al., 2015; Song & Ermon, 2019; Ho et al., 2020) and the DDIM framework (Song et al., 2021) from the interpolation viewpoint (Liu et al., 2023b; Lipman et al., 2023; Albergo et al., 2023). More details are provided in Appendix C.

DDMs define a generative procedure for a data distribution $\pi_0$ by specifying a continuous family of marginals $(\pi_t)_{t \in [0,1]}$ that connects $\pi_0$ to the simple reference distribution $\pi_1 := \mathcal{N}(0, \mathbf{I})$. More precisely, we consider here $\pi_t = \mathrm{Law}(X_t)$, where

$$X_t = \alpha_t X_0 + \sigma_t X_1 \, , \qquad (7)$$

$X_0$ and $X_1$ are independent samples from $\pi_0$ and $\pi_1$ respectively. In addition, $(\alpha_t)_{t \in [0,1]}$ and $(\sigma_t)_{t \in [0,1]}$ are non-increasing and non-decreasing schedules, respectively, with boundary conditions $(\alpha_0, \sigma_0) := (1, 0)$ and $(\alpha_1, \sigma_1) := (0, 1)$. To generate new samples, DDMs simulate a time-reversed Markov chain. Given a decreasing sequence

$(t_k)_{k=0}^{n-1}$ of $n$ time steps with $t_{n-1} = 1$ and $t_0 = 0$, reverse transitions are applied iteratively to map a sample from $\pi_{t_{k+1}}$ to one from $\pi_{t_k}$, progressively denoising until the clean data distribution $\pi_0$ is reached.

The DDIM framework (Song et al., 2021) introduces a general family of reverse transitions for denoising diffusion models. It relies on a schedule $(\eta_t)_{t \in [0,1]}$, satisfying $\eta_t \leq \sigma_t$ for all $t \in [0,1]$, along with a family of conditional distribution given for $s < t$ by

$$q_{s|0,1}^{\eta}(x_s|x_0, x_1) := \mathrm{N}(x_s; \alpha_s x_0 + \sqrt{\sigma_s^2 - \eta_s^2}\, x_1, \eta_s^2 \mathbf{I}) .$$

When $\eta_s = 0$, this Gaussian is understood, by abuse of notation, as a Dirac mass centered at the same mean. Clearly, for all $\eta_s \in [0, \sigma_s]$, a sample from $q_{s|0,1}^{\eta}(\cdot|X_0, X_1)$ with $(X_0, X_1) \sim \pi_0 \otimes \mathcal{N}(0, \mathbf{I})$ is a sample from $\pi_s$. Note that if $X_s^{\eta}|X_0, X_1 \sim q_{s|0,1}^{\eta}(\cdot|X_0, X_1)$, then $X_s^{\eta}|X_0, X_t \sim q_{s|0,t}^{\eta}(\cdot|x_0, x_t) = q_{s|0,1}^{\eta}(\cdot|X_0, (X_t - \alpha_t X_0)/\sigma_t)$ where the joint distribution of the random variables $(X_0, X_t, X_1)$ is defined in (7). We define the reverse transition

$$\pi_{s|t}^{\eta}(x_s|x_t) = \mathbb{E}\left[ q_{s|0,t}^{\eta}(x_s|X_0, X_t) \,\middle|\, X_t = x_t \right] . \quad (8)$$

By construction, the transitions (8) satisfy the marginalization property, *i.e.* for any $0 \leq s < t \leq 1$, $\pi_s(x_s) = \int \pi_{s|t}^{\eta}(x_s|x_t) \pi_t(x_t) \mathrm{d}x_t$. Thus, $(\pi_{t_k|t_{k+1}}^{\eta})_{k=0}^{n-2}$ defines a set of reverse transitions that enable stepwise sampling from the sequence $(\pi_{t_k})_{k=0}^{n-1}$. In practice, however, these transitions are intractable. A common approximation is to replace $X_0$ in the second line of (8) by its conditional expectations (Ho et al., 2020; Song et al., 2021). More precisely, let $\hat{x}_0(x_t, t) := \int x_0 \pi_{0|t}(x_0|x_t) \mathrm{d}x_0$, where $\pi_{0|t}$ is defined as the conditional distribution of $X_0$ given $X_t$ in (7). Then the model proposed in (Ho et al., 2020; Song et al., 2021) corresponds to approximating each $\pi_{t_k|t_{k+1}}^{\eta}$ by

$$\hat{p}_{k|k+1}^{\eta}(x_k|x_{k+1}) := q_{t_k|0,t_{k+1}}^{\eta}(x_k|\hat{x}_0(x_{k+1}, t_{k+1}), x_{k+1}).$$

For simplicity, we consider next only the deterministic sampler with $\eta_s = 0$ for all $s \in [0,1]$. Then $\hat{p}_{k|k+1}^{\eta}(\cdot|x_{k+1})$ becomes a Dirac at $T_{t_k|t_{k+1}}(x_{t_{k+1}})$ where for $s < t$:

$$T_{s|t}(x_t) := (\alpha_s - \alpha_t \sigma_s/\sigma_t)\hat{x}_0(x_t, t) + \sigma_s x_t/\sigma_t . \quad (9)$$

Finally, define for all $k < n - 2$ and $x_1 \in \mathbb{R}^{L \times K}$ the DDIM mapping:

$$T_{t_k}(x_1) := T_{t_k|t_{k+1}} \circ \ldots \circ T_{t_{n-2}|t_{n-1}}(x_1) . \quad (10)$$

When the denoiser $(t, x) \mapsto \hat{x}_0(x, t)$ is intractable it is replaced with a parametric model trained with a denoising loss.

---

**Algorithm 1** Soft reparameterization with DDIM transitions

1: **Input:** grid $(t_k)_{k=0}^{n-1}$, schedule $(\alpha_{t_k}, \sigma_{t_k})_{k=0}^{n-1}$
2: Sample $x \sim \mathcal{N}(0, \mathbf{I}_K)^{\otimes L}$
3: **for** $k = n - 1$ **down to** $0$ **do**
4:     $\hat{x}_0 \leftarrow \mathrm{softmax}(\varphi_\theta + \alpha_{t_{k+1}} x/\sigma_{t_{k+1}}^2)$
5:     $\hat{x}_1 \leftarrow (x^i - \alpha_{t_{k+1}}\hat{x}_0)/\sigma_{t_{k+1}}$
6:     $x \leftarrow \alpha_{t_k}\hat{x}_0 + \sigma_{t_k}\hat{x}_1$
7: **end for**
8: **return** $x$

---

### 3.2. Diffusion-based categorical reparameterization

We now introduce our diffusion-based soft reparameterization of $\pi_\theta$. This reparameterization is based on a DDM with target $\pi_0^\theta = \pi_\theta$. Since $\pi_\theta$ is a discrete measure, the resulting denoising distribution, denoted by $\pi_{0|t}^\theta$, is also discrete. Indeed, by (7) and the factorization (2), the conditional distribution factorizes as $\pi_{0|t}^\theta(x_0|x_t) \propto \prod_{i=1}^L \pi_{0|t}^{\theta,i}(x_0^i|x_t^i)$, where

$$\pi_{0|t}^{\theta,i}(x_0^i|x_t^i) \propto \pi_\theta^i(x_0^i)\mathrm{N}(x_t^i; \alpha_t x_0^i, \sigma_t^2 \mathbf{I}_K) .$$

With this structure, the posterior-mean denoiser $\hat{x}_0^\theta(x_t, t) := \sum_{x_0} x_0 \pi_{0|t}^\theta(x_0|x_t)$ simplifies to a matrix of posterior probabilities due to the one-hot structure; that is, for any $i \in [L]$ and $j \in [K]$, we have $\hat{x}_0^\theta(x_t, t)^{ij} = \pi_{0|t}^{\theta,i}(e_j|x_t)$, and the denoiser can be computed exactly and efficiently. Indeed, since $\|x_t^i - \alpha_t e_j\|^2 = \|x_t^i\|^2 - 2\alpha_t x_t^{ij} + \alpha_t^2$, we get

$$\hat{x}_0^\theta(x_t, t)^{ij} = \frac{\pi_\theta^i(e_j) \exp(-\frac{\|x_t^i\|^2 - 2\alpha_t x_t^{ij} + \alpha_t^2}{2\sigma_t^2})}{\sum_{k=1}^K \pi_\theta^i(e_k) \exp(-\frac{\|x_t^i\|^2 - 2\alpha_t x_t^{ik} + \alpha_t^2}{2\sigma_t^2})}$$

$$= \frac{\exp(\varphi_\theta^{ij}) \exp(\alpha_t x_t^{ij}/\sigma_t^2)}{\sum_{k=1}^K \exp(\varphi_\theta^{ik}) \exp(\alpha_t x_t^{ik}/\sigma_t^2))} .$$

This yields the following simple matrix form for the denoiser:

$$\hat{x}_0^\theta(x_t, t) = \mathrm{softmax}(\varphi_\theta + \alpha_t x_t/\sigma_t^2) . \quad (11)$$

Unlike standard diffusion models that learn an approximate denoiser using a neural network, here the denoiser $\hat{x}_0^\theta(\cdot, t)$ has a closed-form expression due to the factorized categorical structure. This enables reverse transitions from $\pi_1$ to $\pi_\theta$ without denoiser approximation and yields an approximate, differentiable sampling procedure. Denote for any $k < n - 2$ by $T_{t_k}^\theta$ the DDIM map associated with $\hat{x}_0^\theta(x_t, t)$ defined in (9). Then, $T_{t_k}^\theta(X_1)$ with $X_1 \sim \mathcal{N}(0, \mathbf{I}_K)^{\otimes L}$ is an approximate sample from the Gaussian mixture with density $\pi_{t_k}^\theta(x_{t_k}) := \sum_{x_0} \prod_{i=1}^L \mathrm{N}(x_{t_k}^i; \alpha_{t_k} x_0^i, \sigma_{t_k}^2 \mathbf{I}_K)\pi_\theta(x_0)$ and $T_0^\theta(X_1)$ is an approximate relaxed sample from $\pi_\theta$. By (6), a natural choice of gradient estimator is

$$\mathrm{J}_\theta T_0^\theta(X_1)^\top \nabla_x f(X_0) \,, \qquad \text{(REDGE)}$$

where $X_0 \sim \pi_{0|t_1}^\theta(\cdot | T_{t_1}^\theta(X_1))$. As a result, with a single diffusion step, the reparameterized sample is $T_0^\theta(X_1) = \hat{x}_0^\theta(X_1, 1) = \mathbb{E}_{\pi_\theta}[X_0]$, due to the boundary condition $\alpha_1 = 0$, and we recover the STRAIGHT-THROUGH estimator (both soft and hard) as a special case. In contrast, using many diffusion steps and appropriately chosen timesteps $(t_k)_{k=0}^{n-1}$ yields an almost exact reparameterization of $\pi_\theta$. As discussed previously, this is precisely the regime we seek to avoid: the gradient of the mapping essentially vanishes, resulting in a high-variance reparameterized gradient. This trade-off is directly analogous to the role of the temperature parameter $\tau$ in GUMBEL-SOFTMAX relaxations, where a high temperature yields a relaxed but biased approximation, while a low temperature results in a high-variance estimator. In our case, the relaxation parameter is determined by the number of diffusion steps and the placement of the timesteps $(t_k)_{k=1}^{n-1}$.

More precisely, Proposition 1 characterizes, at a fixed number of timesteps, the behavior of the reparameterized gradient as $t_1 \to 0$. The proof is given in Appendix A.

**Proposition 1.** *With $L = 1$ and the timesteps $(t_k)_{k=2}^{n-1}$ fixed, under assumptions stated in the Appendix, we have for all $\theta \in \Theta$,*

$$\lim_{t_1 \to 0} \left\| \mathrm{J}_\theta T_0^\theta(X_1) \right\| = 0 \,, \quad \mathbb{P}\text{-a.s.} \qquad (12)$$

*with $X_1 \sim \mathcal{N}(0, \mathbf{I}_K)$.*

The proof consists in showing that, as $t_1 \to 0$, the last DDIM step $T_{0|t_1}^\theta$ collapses almost all points in $\mathbb{R}^K$ onto a single one-hot vector, and as a consequence, the Jacobian of $T_0^\theta$ with respect to $\theta$ vanishes. We illustrate Proposition 1 in Figure 1. Following the previous discussion, $t_1$ should not be chosen so small that the gradients become uninformative.

### 3.3. Extensions

**REINMAX extension.** We derive a REINMAX (Liu et al., 2023a) version of our diffusion-based reparameterization trick. First, by the marginalization property we have that $\pi_\theta(x_0) = \int \pi_{0|t_1}^\theta(x_0|x_{t_1})\pi_{t_1}^\theta(x_{t_1})\mathrm{d}x_{t_1}$, and we can write, using the tower property, that $\mathbb{E}_{\pi_\theta}[f(X)] = \mathbb{E}[h_\theta(X_{t_1})]$, where $X_{t_1}$ is given by (7) with $\pi_0 = \pi_\theta$ and $h_\theta(x_{t_1}) := \sum_{x_0} f(x_0) \pi_{0|t_1}^\theta(x_0|x_{t_1})$. The Gaussian mixture $\pi_{t_1}^\theta$ can be reparameterized approximately using the DDIM map $T_{t_1}^\theta$ in (10) and therefore $\mathbb{E}[h_\theta(X_{t_1})] \approx \mathbb{E}[h_\theta(T_{t_1}^\theta(X_1))]$ for any $\theta \in \Theta$. A Monte Carlo estimator of the total gradient of the r.h.s. at $\theta = \theta'$ is given by

$$\nabla_\theta h_\theta(T_{t_1}^{\theta'}(X_1))\big|_{\theta=\theta'} + \mathrm{J}_\theta T_{t_1}^\theta(X_1)^\top\big|_{\theta=\theta'} \nabla_x h_{\theta'}(T_{t_1}^{\theta'}(X_1)),$$

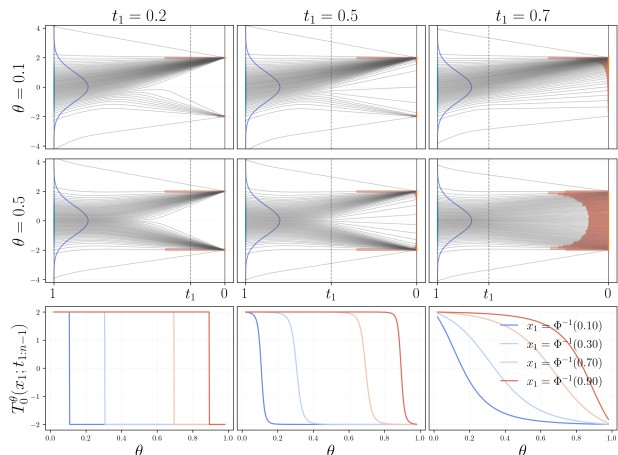

*Figure 1.* Visualization of the DDIM transport for $\pi_\theta = \theta\cdot\delta_{-2} + (1-\theta)\cdot\delta_2$ with the linear schedule $(\alpha_t, \sigma_t) = (1-t, t)$. **First two rows**: DDIM trajectories with varying $t_1$ for two different values of $\theta \in [0,1]$. **Third row**: The DDIM map $\theta \mapsto T_0^\theta(x_1; t_{1:n-1})$ for fixed input quantiles $z$ and three different values of $t_1$. $\Phi$ stands for the standard Gaussian cdf.

where the intractable terms are the gradients w.r.t. $\theta$ and $x$ of the conditional expectation $h_\theta$. The key observation is that for any $x_{t_1}$, the gradient of $\theta \mapsto h_\theta(x_{t_1})$ is a specific case of differentiating an expectation w.r.t. the parameters of a categorical distribution, which in this case is $\pi_{0|t_1}^\theta(\cdot|x_{t_1})$. Here by using the STRAIGHT-THROUGH approximation (4) we recover our hard gradient estimator (REDGE); *i.e.* $\nabla_\theta h_\theta(x_{t_1}) \approx \nabla_\theta f(\hat{x}_0^\theta(x_{t_1}, t_1))$. Our REINMAX-based estimator replaces hard ST with REINMAX (5) as an estimator of $\nabla_\theta h_\theta(x_{t_1})$. We refer to this gradient estimator as REINDGE. When using a single diffusion step, *i.e.* $t_1 = 1$, the function $h_\theta$ is constant and equal to $\mathbb{E}_{\pi_\theta}[X]$ due to the boundary condition $\alpha_1 = 0$, and REINMAX is recovered as a special case. The same observation holds for the map $x_{t_1} \mapsto h_\theta(x_{t_1})$, for any $\theta$, but here we simply use the STRAIGHT-THROUGH estimator.

**Parameter dependent $\pi_1$.** In the previous construction, the terminal distribution $\pi_1$ is fixed to a standard Gaussian $\pi_1 = \mathcal{N}(0, \mathbf{I}_K)^{\otimes L}$. In our setting, however, we can exploit the factorization (2) to select a *parameter–dependent* Gaussian distribution $\pi_1^\theta$ that best approximates $\pi_\theta$ in the maximum–likelihood sense. Specifically, we take $\pi_1^\theta$ with factorized density $\pi_1^\theta(x) = \prod_{i=1}^L \mathrm{N}(x^i; \mu_\theta^i, \mathrm{Diag}(v_\theta^i))$, where for all $i \in [L]$, $(\mu_\theta^i, v_\theta^i) \in \mathbb{R}^K \times \mathbb{R}_{>0}^K$ and $\mathrm{Diag}(v_\theta^i) \in \mathbb{R}^{K \times K}$ is a diagonal matrix with $v_\theta^i$ as diagonal entries. The parameters are then defined as any solution to the maximum–likelihood problem of maximizing $\mathbb{E}_{\pi_\theta}[\log \pi_1^\theta(X_0)]$ w.r.t. $(\mu_\theta, v_\theta)$ whose one solution is given by matching the mean and per–coordinate variances of $\pi_\theta^i$; *i.e.* $\mu_\theta^i = \mathbb{E}_{\pi_\theta^i}[X_0^i]$ and $v_\theta^i = \mu_\theta^i \odot (1 - \mu_\theta^i)$. We restrict ourselves to a diagonal covariance in order to avoid expensive matrix

inversions in the denoiser expression derived next. Data–dependent base distributions of this kind have also been considered in other applications, see for instance gil Lee et al. (2022); Popov et al. (2021); Luo et al. (2023); Ohayon et al. (2025). When using the base distribution $\pi_1^\theta$ and setting $\eta_s = 0$ for all $s \in [0, 1]$, the DDIM map (9) keeps the same form as before. The denoiser, however, is different and is now given in matrix form by

$$\hat{x}_0^\theta(x_t, t) = \text{softmax}(\varphi_\theta + \frac{\alpha_t \lambda_\theta}{\sigma_t^2} \odot (x_t - \sigma_t \mu_\theta - \frac{\alpha_t}{2}\mathbf{1})). \tag{13}$$

where $\lambda_\theta \in \mathbb{R}^{L \times K}$ with $\lambda_\theta^{i,j} = 1/v_\theta^{i,j}$ and $\mathbf{1} \in \mathbb{R}^{L \times K}$ is the all-ones matrix. See Appendix C.2 for a derivation and Appendix C for the DDIM sampler with arbitrary schedule $(\eta_s)_{s \in [0,1]}$. We refer to the resulting gradient estimator as REDGE-COV. Finally, for large vocabularies $K$, the diagonal covariance can become ill-conditioned, so we also consider a scalar variant with $\text{Diag}(v_\theta^i) = \sigma_\theta^2 \mathbf{I}_K$.

### 3.4. Related work

**Reparameterization trick.** Beyond GUMBEL-SOFTMAX, several works propose alternative approximate reparameterizations. Potapczynski et al. (2020) replace Gumbel noise with an invertible push-forward of a Gaussian, yielding a richer family of simplex-valued relaxations. Wang & Yin (2020) relax the factorization assumption in (2) by modeling correlated multivariate Bernoulli variables via a Gaussian copula. Paulus et al. (2020a) generalize the Gumbel–max trick through solutions of random linear programs, obtaining differentiable relaxations by adding a strongly convex regularizer.

**Denoiser for a mixture of Dirac deltas.** Dieleman et al. (2022) propose CDCD, which models categorical data by training a diffusion model on embedded tokens. Because the underlying variables are discrete, the denoiser is learned with a cross-entropy objective. When fitting a diffusion model to a finite dataset $(X_i)_{i=1}^N$, the minimizer of the denoising objective is precisely the denoiser associated with the empirical distribution $N^{-1} \sum_{i=1}^N \delta_{X_i}$, and it admits a closed-form expression; see Karras et al. (2022, Appendix B.3). Since our purpose is not to synthesise new data but to have differentiable sampling procedure, we similarly exploit closed-form denoisers for distributions on $V^L$. In concurrent work, Andersson & Zhao (2025) propose using diffusion models within a sequential Monte Carlo setting to generate $N$ i.i.d. reparameterized samples from the parameter-dependent empirical mixture $\sum_{i=1}^N w_i^\theta \delta_{X_i^\theta}$, where $w_i^\theta \geq 0$ and $\sum_{i=1}^N w_i^\theta = 1$, and $\theta$ denotes the state-space model parameters. This enables parameter estimation by differentiating end-to-end through the particle filter used to estimate the observation likelihood.

## 4. Experiments

In this section, we evaluate our method on benchmark problems spanning Sudoku solving and generative modeling. Further experiments are given in Appendix E and F. We compare against three representative baselines: the STRAIGHT-THROUGH (ST) estimator (Bengio et al., 2013), GUMBEL-SOFTMAX (using its straight-through variant) (Jang et al., 2017), and REINMAX (Liu et al., 2023a). We focus on these since REINMAX is a recent strong method that reports state-of-the-art results and is shown to outperform several earlier alternatives (Liu et al., 2023a), so we omit additional baselines. In addition, we don't compare against REBAR/RELAX-style estimators (Tucker et al., 2017; Grathwohl et al., 2018) because they are meta-estimators that wrap a base estimator with learned control variates and additional tuning. Our method could in principle be used as the base reparameterization within these frameworks, which we leave to future work.

All hyperparameters are reported in Appendix G.1. We also report the runtime and memory usage in Appendix G.4. For all methods we use the hard version. For REDGE and its variants, we use the linear schedule $(\alpha_t, \sigma_t) = (1 - t, t)$ (Lipman et al., 2023; Esser et al., 2024). For the timesteps we first specify $t_1$ and then set $t_k = t_1 + (1 - t_1)k/(n-1)$ for $k \in [2 : n - 1]$.

### 4.1. Inference-time guidance with Masked Diffusion

We start by providing some necessary background on Masked Diffusion models (MDM) (Shi et al., 2024; Sahoo et al., 2024). We defer a more formal introduction to Appendix D.

**Masked diffusion.** Let $p$ be a target distribution defined on X with the vocabulary augmented by the mask token m. MDMs provide an approximate sampler for $p$ via an iterative unmasking process. We denote the learned model distribution by $p_0^{d,\psi}$, where $\psi$ are the model parameters. We use the superscript d (for discrete) to avoid conflict with the Gaussian diffusion notation. MDMs rely on a clean-data predictor $p_{0|k}^{d,\psi}(\cdot|X_k)$ that outputs a factorized categorical approximation (2) to the posterior of $X_0 \sim p$ given a partially masked state $X_k$, under a joint distribution where $X_k$ is obtained from $X_0$ by setting independently across the dimensions $X_k^i = $ m with probability $(1 - \beta_k)$ and $X_k^i = X_0^i$ otherwise. $(\beta_k)$ is chosen as a decreasing schedule. Sampling proceeds by simulating a Markov chain $(X_{0:M})$ where $X_M^i = $ m for all $i \in [K]$, and given $X_k$, we first draw an approximate solution $\hat{X}_0 \sim p_{0|k}^{d,\psi}(\cdot|X_k)$, then sample $X_{k-1}$ by keeping the unmasked entries of $X_k$ fixed and, for masked entries, setting $X_{k-1}^i = \hat{X}_0^i$ with probability $(\beta_{k-1} - \beta_k)/(1 - \beta_k)$ (otherwise $X_{k-1}^i = $ m).

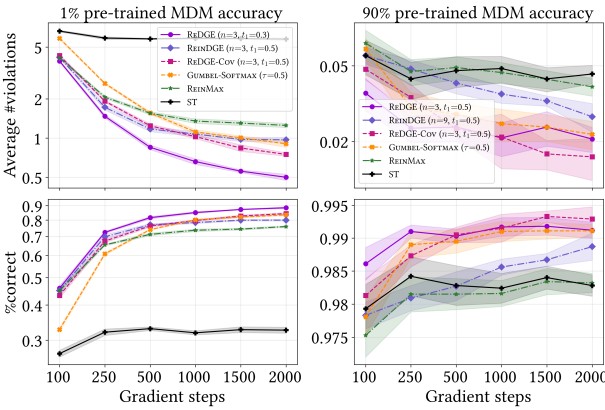

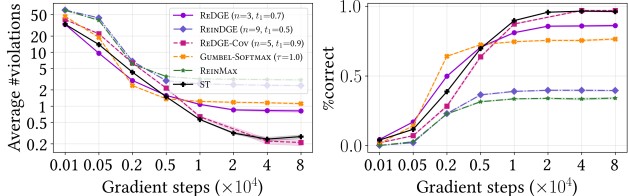

*Figure 2.* Masked diffusion guidance on Sudoku: fraction solved and mean constraint violations (1000 test puzzles, 10 seeds, 20 diffusion steps), for early (1%) and late (90%) checkpoints, as a function of the gradient-step budget. For each estimator, we sweep hyperparameters and learning rates, select the setting that minimizes the AUC of mean violations over step budgets (100–2000), and plot its violations and solve rate across budgets.

**Inference-time guidance.** Given a reward $r$ we want to steer sampling at test time by locally modifying the model's step-wise predictive distribution to favor samples with higher reward. Following (Murata et al., 2025), this can be achieved by training, given $x_k$ at diffusion step $k$, a factorized variational distribution $\pi_\theta(\cdot|x_k)$ to approximate the *tilted* distribution with p.m.f. at $x_0$ proportional to $\exp(-r(x_0))p_{0|k}^{\mathsf{d},\psi}(x_0|x_k)$. This is done by minimizing the forward KL divergence objective

$$F_k(\theta) := \mathbb{E}_{\pi_\theta(\cdot|x_k)}[r(X_0)] + \mathrm{KL}(\pi_\theta(\cdot|x_k)\|p_{0|k}^{\mathsf{d},\psi}(\cdot|x_k)) \,. \tag{14}$$

We then draw $\hat{X}_0$ from the obtained proposal and then sample $X_{k-1}$ as previously done. We provide more details in Appendix D. We consider two such applications in the next subsections. In all cases, we optimize the logits directly by setting $\varphi_\theta = \theta$ and treating $\theta \in \mathbb{R}^{L \times K}$ as the optimization variable. We detail the guidance algorithm in Appendix D.

### 4.1.1. MDM GUIDANCE FOR SOLVING SUDOKU PUZZLES.

We follow Ye et al. (2024) and train a masked diffusion model (MDM) to approximate the distribution $p(\cdot|\mathbf{c})$ over valid completions of an incomplete Sudoku grid $\mathbf{c}$, viewed as a categorical distribution on $\mathsf{V}^{81}$, where $\mathsf{V}$ denotes the set of one-hot vectors of length 10 and the mask $\mathsf{m}$ is $e_{10}$. Let $\mathcal{G}$ denote the 27 constraint groups (rows, columns, and blocks). For $g \in \mathcal{G}$, define the digit-count map $s_g(X) := \sum_{i \in g} PX^i$, where $P$ drops the mask coordinate so that $s_g(X) \in \mathbb{R}^9$ counts digits in $g$. We use the reward $r(x) := \sum_{g \in \mathcal{G}} \|s_g(x) - \mathbf{1}_9\|_2^2$. For the first experiment, we use two checkpoints with very different baseline performance: an

*Figure 3.* Solving Sudoku without pre-trained MDM. Similarly to Figure 2 we select a single configuration by minimizing the area under the mean-violation curve over budgets (100 to 8e4 steps)

early model that solves about $1\%$ of the 1000 test Sudokus and a late model that solves $90\%$. We apply inference-time guidance by optimizing (14), estimating the reward-gradient term with hard gradient estimators and differentiating the KL term exactly.

*Results.* The results are given in Figure 2. From the 1% checkpoint, guidance with REDGE raises the solve rate to 89%, outperforming the strongest baselines, which plateau in the mid-80s. STRAIGHT-THROUGH performs substantially worse, and using a smaller $t_1$ (REINDGE) improves over the larger-$t_1$ REINMAX special case. Starting from the 90% checkpoint, guidance further improves performance, reaching solve rate 93% with REDGE-COV.

**Direct optimization without pre-training.** We also study a no-prior variant in which we drop the MDM entirely and directly optimize $\mathbb{E}_{\pi_\theta}[r(X)]$ w.r.t. the parameters of factorized categorical distribution $\pi_\theta$. The Sudoku clues in $\mathbf{c}$ are enforced by setting the logit of the observed digit to a very large value at each clue location after every gradient step (whereas with a pre-trained MDM this conditioning is already reflected in the posterior initialization). Surprisingly, the best-performing estimators achieve solve rates in the mid-to-high 90s, substantially higher than what is obtained when guiding from the weak 1% checkpoint, while REDGE and GUMBEL-SOFTMAX lag behind. Finally, we provide a comprehensive heatmap summarizing how the performance of REDGE and REDGE-COV behaves as a function of $(n, t_1)$. We make three empirical remarks; (i) we can see that in all cases a smaller $t_1$ results in worst performance, as suggested by our theoretical analysis and Figure 1. (ii) Despite its strong performance REDGE-COV is more sensitive to the hyperparameters than REDGE. (iii) Using more diffusion steps doesn't affect the performance much, except for very small $t_1$. This connection between $t_1$, $n$ and the performance of our gradient estimators is discussed and studied in more detail in Appendix E.2.

### 4.1.2. REWARD-GUIDED IMAGE GENERATION

We next apply inference-time guidance to discrete image generation with a class-conditional pretrained MaskGIT

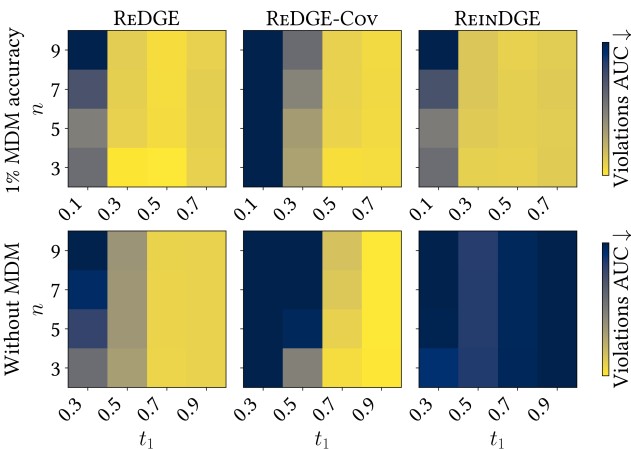

*Figure 4.* Average violations heatmap as a function of $n$ and the timestep $t_1$. For each configuration $(n, t_1)$ we report the lowest AUC obtained over a sweep of four learning rates.

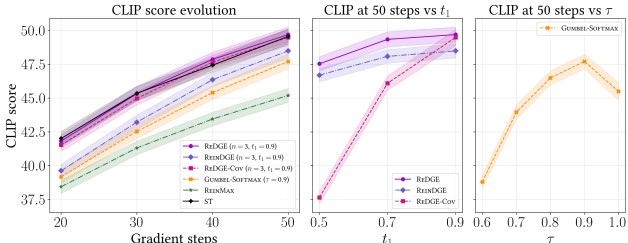

*Figure 5.* Left: Average CLIP score for CLIP-guided image generation. Middle and right: sensitivity of CLIP score to the temperature parameters $t_1$ and $\tau$. We report the mean over 200 images; for each estimator, we sweep hyperparameters and learning rates, select the setting that maximizes the AUC of CLIP score over gradient steps budget.

(Chang et al., 2022) model trained on the ImageNet dataset and operating on VQ-VAE codes. We generate images at resolution $384 \times 384 \times 3$ (Besnier et al., 2025) by sampling a sequence of discrete latent codes $[K]^L$, where each image is represented by $L = 576$ codes, each taking one of $K = 16384$ codebook entries. Each image is thus represented by a latent embedding in $\mathbb{R}^{576 \times d}$ with $d = 8$. We write $E$ for the embedding matrix in $\mathbb{R}^{K \times d}$. Given $x \in \mathsf{X}$, the reward consists in decoding the embedding $x \cdot E \in \mathbb{R}^{L \times d}$ and then computing the CLIP score (Radford et al., 2021; Hessel et al., 2021) with a target prompt.

*Results.* As shown in Figure 5, guidance monotonically improves CLIP score as a function of the gradient step budget. REDGE, REDGE-COV, and STRAIGHT-THROUGH achieve comparable performance, outperforming REINDGE and GUMBEL-SOFTMAX and substantially surpassing REINMAX. The middle panel of Figure 5 shows that performance peaks at $t_1 = 0.9$, a regime that is closer to straight-through behavior and is consistent with STRAIGHT-THROUGH also performing well. However, strong perfor-

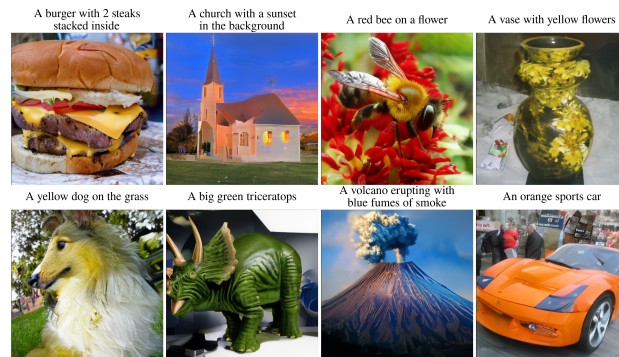

*Figure 6.* REDGE samples generated by CLIP-guided MaskGIT from the prompts shown.

mance persists for $t_1 \in \{0.5, 0.7\}$, indicating a broad operating range for REDGE and suggesting that its gains are not solely driven by the near-STRAIGHT-THROUGH limit. On the other hand, REDGE-COV and GUMBEL-SOFTMAX exhibit substantially higher hyperparameter sensitivity.

**Takeaways and practical tuning.** Across our benchmarks, REDGE is the most reliable, delivering strong performance across diverse objectives, whereas our variants as well as STRAIGHT-THROUGH, REINMAX, and GUMBEL-SOFTMAX are more setting-dependent. REDGE-COV can be particularly strong in favorable regimes but is more hyperparameter-sensitive than REDGE, which offers the best robustness–performance trade-off. For tuning, a small number of diffusion steps (e.g., $n = 3, 5$) coupled with a moderate $t_1$ (e.g. $t_1 \in \{0.5, 0.7, 0.9\}$) is a strong default. The endpoint $t_1 = 1$ recovers the straight-through limit and is a useful reference when STRAIGHT-THROUGH is competitive, yet strong results often persist for intermediate $t_1$, indicating gains beyond the near-STRAIGHT-THROUGH regime.

## 5. Conclusion

We introduced REDGE, a diffusion-based approach to categorical reparameterization that leverages the fact that, for categorical distributions, the denoiser is available in closed form, yielding a training-free differentiable sampling map from Gaussian noise to $\pi_\theta$. We analyzed the effect of $t_1$ (playing the role of a temperature) and explained how near-constant transport regions and sharp decision boundaries arise as the relaxation tightens, leading to uninformative gradients. The resulting family of estimators includes hard variants and recovers STRAIGHT-THROUGH and REIN-MAX as one-step special cases. Beyond improving default schedules and diagnostics for robust hyperparameter selection, a promising direction is to reduce residual bias using REBAR/RELAX-style control variates, treating REDGE as a strong base pathwise estimator.

## Acknowledgments

This work was partially supported by Digital Futures through the call "Strengthening French-Swedish AI Collaboration." J. Olsson was also supported by the Swedish Research Council under grant 2024-05680.

## Impact Statement

This paper presents work whose goal is to advance the field of machine learning. There are many potential societal consequences of our work, none of which we feel must be specifically highlighted here.

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

# A. Proofs

## A.1. Gradient instability: statement of Proposition 1 and its conditions

In this section we assume without loss of generality that $L = 1$, $K \geq 2$, and $\varphi_\theta = \theta \in \mathbb{R}^K$. We also define for all $t \in (0, 1]$, $c_t = \alpha_t / \sigma_t^2$. With these notations, noting that $\hat{x}_0^\theta(x, t)$ is the probability vector associated to $\pi_{0|t}^\theta(\cdot | x)$:

$$\hat{x}_0^\theta(x, t) := \text{softmax}(\theta + c_t x) , \quad x \in \mathbb{R}^K , \quad \hat{x}_0^\theta(x, t)^i = \frac{\exp(\theta^i + c_t x^i)}{\sum_{k=1}^K \exp(\theta^k + c_t x^k)} = \pi_{0|t}^\theta(e_i | x) , \quad i \in \{1, \ldots, K\} . \quad (15)$$

In addition, recall the notation

$$\Sigma_t^\theta(x) := \mathbb{C}\text{ov}_{\pi_{0|t}^\theta(\cdot | x)}(X) . \quad (16)$$

Finally, define the union of decision boundaries:

$$\mathsf{H} := \{x \in \mathbb{R}^K : \text{ there exists } j, k \in [K], x^j = x^k = \max_i x^i\} . \quad (17)$$

We define the margin function $m : \mathbb{R}^K \to \mathbb{R}$ as the gap between the largest and second-largest coordinates

$$m(x) := \max_i x^i - m_2(x) , \quad m_2(x) = \begin{cases} \max\{x^j : j \in \{1, \ldots, K\}, x^j \neq \max_i x^i\} & \text{if there exists } x^j \neq \max_i x^i \\ \max_i x^i & \text{otherwise} . \end{cases} \quad (18)$$

Note that for $x \notin \mathsf{H}$, $\text{argmax}_{j \in [K]} x^j$ is reduced to a singleton and therefore,

$$m(x) := \min_{j \neq k^*(x)} (x^{k^*(x)} - x^j) , \quad k^*(x) = \underset{j \in [K]}{\text{argmax}} \, x^j . \quad (19)$$

We now consider the following assumptions.

**(A1)** The schedule $(\alpha_t, \sigma_t)_{t \in [0,1]}$ is such that $\lim_{t \to 0} c_t = \infty$ where we recall that $c_t = \alpha_t / \sigma_t^2$.

**Proposition 2.** *Fix $\theta \in \mathbb{R}^K$ and suppose that (A1) holds. Consider the DDIM sampler $T_0^\theta : \mathbb{R}^K \to \mathbb{R}^K$ with the last time step $t_1 \in (0, 1)$ and all other time steps $(t_k)_{k \geq 2}$ fixed. Then, for any $x_1 \in \mathbb{R}^K$ such that $T_{t_1}^\theta(x_1) \notin \mathsf{H}$, there exists $M(t_t) \geq 0$ only depending on $t_2$ such that*

$$\left\| \mathsf{J}_\theta T_0^\theta(x_1) \right\| \leq 2K(K-1)(1 + c_{t_1} M(t_2)) \exp\left( -m(T_{t_1}^\theta(x_1)) c_{t_1} / 2 \right) . \quad (20)$$

Consider now the additional assumption:

**(A2)** For any $\theta \in \mathbb{R}^K$, there exists a measurable map $\widetilde{X}_0^\theta : \mathbb{R}^K \to \mathbb{R}^K$ such that for $X_1 \sim \mathcal{N}(0, \mathbf{I}_K)$, $\mathbb{P}$-almost surely it holds

$$\lim_{t_1 \to 0} T_{t_1}^\theta(X_1) = \widetilde{X}_0^\theta(X_1) \quad \text{and} \quad \widetilde{X}_0^\theta(X_1) \notin \mathsf{H} .$$

Assumption (A2) is a mild local regularity and non-degeneracy assumption on the DDIM sampler with $t_1$ near 0. In particular, it the number of DDIM step is equal to 1, it easy to verify that $\lim_{t_1 \to 0} T_{t_1}^\theta(X_1)$ converges to the one-hot vector associated to $\text{argmax}_i X^i$ and therefore (A2) holds. Furthermore, (A2) only requires that, for each $\theta \in \mathbb{R}^K$, the trajectory $t_1 \mapsto T_{t_1}^\theta(X_1)$, started from Gaussian noise $X_1 \sim \mathcal{N}(0, \mathbf{I}_K)$, admits an almost-sure limit as $t_1 \to 0$, and that this limit does not lie on the decision boundary $\mathsf{H}$. In particular, we do *not* assume that $\widetilde{X}_0^\theta(X_1)$ coincides with the data distribution or that it is one-hot; we only use that the limiting state is well-defined and is not in $\mathsf{H}$.

**Corollary 1.** *Fix $\theta \in \mathbb{R}^K$ and suppose that (A1)-(A2) hold. Let $X_1 \sim \mathcal{N}(0, \mathbf{I}_K)$ and consider the DDIM sampler $T_0^\theta : \mathbb{R}^K \to \mathbb{R}^K$ with the last time step $t_1 \in (0, 1)$ and all other time steps $(t_k)_{k \geq 2}$ fixed. Then, $\mathbb{P}$-almost surely*

$$\lim_{t_1 \to 0} \left\| \mathsf{J}_\theta T_0^\theta(X_1) \right\| = 0 . \quad (21)$$

In the next section, we state and prove preliminary results needed for the proof of Proposition 2 postponed to Appendix A.3.

## A.2. Supporting Lemmas for Proposition 1

**Lemma 1.** *For each $t \in (0,1]$ and $x \in \mathbb{R}^K$,*

$$\mathrm{J}_\theta \hat{x}_0^\theta(x,t) = \Sigma_t^\theta(x) , \quad \mathrm{J}_x \hat{x}_0^\theta(x,t) = c_t \Sigma_t^\theta(x) .$$

*Proof.* By (15), a direct computation gives, for all $i, j \in [K]$, $x, \theta$,

$$\partial_{\theta^j} \hat{x}_0^\theta(x,t)^i = \hat{x}_0^\theta(x,t)^i (\delta_{ij} - \hat{x}_0^\theta(x,t)^j) ,$$

so in matrix form

$$\mathrm{J}_\theta \hat{x}_0^\theta(x,t) = \mathrm{Diag}\big(\hat{x}_0^\theta(x,t)\big) - \hat{x}_0^\theta(x,t)\hat{x}_0^\theta(x,t)^\top .$$

By definition, $\Sigma_t^\theta(x) = \mathbb{E}_{\pi_{0|t}^\theta(\cdot|x)}[X_0 X_0^\top] - \hat{x}_0^\theta(x,t)\hat{x}_0^\theta(x,t)^\top$, where $(X_0, X_t)$ follows the distribution with density $\pi_\theta(x_0)\mathrm{N}(x_t; \alpha_t x_0, \sigma_t^2 \mathbf{I}_K)$, and by (15)

$$\mathbb{E}_{\pi_{0|t}^\theta(\cdot|x)}[X_0 X_0^\top] = \sum_{i=1}^K e_i e_i^\top \pi_{0|t}^\theta(e_i|x) = \mathrm{Diag}(\hat{x}_0^\theta(x,t))$$

and hence the equality $\mathrm{J}_\theta \hat{x}_0^\theta(x,t) = \Sigma_t^\theta(x)$. The Jacobian w.r.t. $x$ follows using similar arguments. $\qquad\square$

**Lemma 2** (Continuity of the margin function outside of H (17)). *$m_2$ is continuous on $\mathbb{R}^K \setminus \mathsf{H}$ and therefore $m$ as well.*

*Proof.* Note that $\mathbb{R}^K \setminus \mathsf{H}$ is the disjoint union of the open sets $\mathsf{U}_i = \{x \in \mathsf{H} : i = \mathrm{argmax}_j x^j\}$. Since on $\mathsf{U}_i$, $m_2(x) = \max_{j \neq i} x^j$, we obtain that $m_2$ is continuous on $\mathbb{R}^K \setminus \mathsf{H}$. $\qquad\square$

**Lemma 3** (Softmax bound). *Let $z \notin \mathsf{H}$ where $\mathsf{H}$ is defined in (17) and $p(z) := \mathrm{softmax}(z)$. Then,*

$$1 - p(z)^{k^*(z)} \le (K-1) \exp\big(-m(z)\big) , \tag{22}$$

*and for all $j \neq k^*(z)$,*

$$p(z)^j \le \exp\big(-m(z)\big) . \tag{23}$$

*Proof.* For ease of notation, we simply denote $p(z)$ by $p$. Since $z \notin \mathsf{H}$, We have that

$$p^j = \frac{\exp(z^j)}{\sum_{\ell=1}^K \exp(z^\ell)} = \frac{\exp(z^j - z^{k^*(z)})}{1 + \sum_{\ell \neq k^*(z)} \exp(z^\ell - z^{k^*(z)})}$$

and for every $j \neq k^\star(z)$, we have $z^{k^\star(z)} - z^j \ge m(z)$, so $z^j - z^{k^\star(z)} \le -m(z)$ and $p^j \le \exp(-m(z))$. Then

$$1 - p^{k^*(z)} = \sum_{j \neq k^*(z)} p^j \le (K-1)\exp(-m(z)) .$$

$\square$

**Lemma 4** (Covariance control). *Let $p \in \Delta^{K-1}$ and $\Sigma = \mathrm{Diag}(p) - pp^\top$. Let $p^{\max} := \max_{j \in [K]} p^j$. Then it holds that*

$$\sum_{j,k=1}^K |\Sigma^{jk}| \le 2K(1 - p_{\max}) .$$

*As a consequence, $\|\Sigma\| \le 2K(1 - p_{\max})$, where $\|\cdot\|$ is the operator norm.*

*Proof.* By definition of the covariance matrix $\Sigma$, we have that $\Sigma^{jj} = p^j(1-p^j)$ and $|\Sigma^{jk}| = p^j p^k$. Let $k^* = \mathrm{argmax}_{i \in [K]} p^i$ and define $p^{\max} = p^{k^*}$. For all $j \in [K]$,

$$\sum_{k=1}^K |\Sigma^{jk}| = \Sigma^{jj} + \sum_{k \neq j} p^j p^k = p^j(1-p^j) + p^j \sum_{k \neq j} p^k = 2p^j(1-p^j) .$$

Next, we have that $p^j(1 - p^j) \leq 1 - p^{\max}$ since if $j = k^*$ then $p^j(1 - p^j) \leq 1 - p^{\max}$ and if $j \neq k^*$ then $p^j(1 - p^j) \leq p^j \leq \sum_{\ell \neq k^*} p^\ell = 1 - p^{\max}$. Hence

$$\sum_{k=1}^{K} |\Sigma^{jk}| \leq 2(1 - p^{\max}) .$$

The final bound is an easy consequence of the norm equivalent in finite dimension. □

We define the notation $a(s,t) = \alpha_s - \alpha_t \sigma_s / \sigma_t$ and $b(s,t) = \sigma_s / \sigma_t$ so that the one-step map writes

$$T_{s|t}^\theta(x) = a(s,t)\, \hat{x}_0^\theta(x,t) + b(s,t)x . \tag{24}$$

**Lemma 5** (DDIM Jacobian bound). *There exists a finite constant $M(t_2) < \infty$, depending only on $t_2$, $K$ and the schedule $(\alpha_t, \sigma_t)$, such that for all $x_1 \in \mathbb{R}^K$ and all $t_1 \in (0, t_2)$,*

$$\left\| J_\theta T_{t_1}^\theta(x_1) \right\| \leq M(t_2) . \tag{25}$$

*In particular, the bound in (25) does not depend on $t_1$.*

*Proof.* **Single-step bound.** We start with a single-step bound on the Jacobian of the map $T_{s|t}^\theta$ with $s < t$. For fixed $t \geq t_2$ and $s \in [0, t]$, the reverse step $T_{s|t}^\theta$ has the form (24), so using Lemma 1 so we obtain

$$J_\theta T_{s|t}^\theta(x) = a(s,t)\, \Sigma_t^\theta(x) ,$$
$$J_x T_{s|t}^\theta(x) = a(s,t)\, c_t\, \Sigma_t^\theta(x) + b(s,t)I_K .$$

Since the schedule $t \mapsto (\alpha_t, \sigma_t, 1/\sigma_t)$ is continuous on $[t_2, 1]$, since $t_2 > 0$, the coefficients $a(s,t), b(s,t)$ and $c_t$ are bounded on the compact set $\{(s,t) : 0 \leq s \leq t, t_2 \leq t \leq 1\}$. Therefore, the uniform covariance bound from Lemma 4 implies that there exist finite constants $L_1(t_2), L_2(t_2)$ such that for all $t \in [t_2, 1]$, $s \in [0,t]$, $x \in \mathbb{R}^K$ and $\theta \in \mathbb{R}^K$,

$$\left\| J_\theta T_{s|t}^\theta(x) \right\| \leq L_1(t_2), \qquad \left\| J_x T_{s|t}^\theta(x) \right\| \leq L_2(t_2) . \tag{26}$$

**Bound via induction.** Next, for each $k \in [1 : n-1]$ we use the following notation for the parameter Jacobian

$$G_k(x_1, \theta') := J_\theta T_{t_k}^\theta(x_1)\big|_{\theta = \theta'} .$$

By construction, the initial state at time $t_{n-1} = 1$ does not depend on $\theta$, so $G_{n-1}(x_1, \theta') = 0$ for all $x_1$.

For $k = 2, \ldots, n-1$ we have, by definition of the sampler,

$$T_{t_k}^\theta(x_1) = T_{t_k | t_{k+1}}^\theta\left(T_{t_{k+1}}^\theta(x_1)\right) .$$

Applying the chain rule with respect to $\theta$ at $\theta_0$ gives

$$G_k(x_1, \theta') = J_\theta T_{t_k | t_{k+1}}^\theta\left(T_{t_{k+1}}^{\theta'}(x_1)\right)\big|_{\theta = \theta'} + J_x T_{t_k | t_{k+1}}^{\theta'}\left(T_{t_{k+1}}^{\theta'}(x_1)\right) \cdot G_{k+1}(x_1, \theta') .$$

We now show by induction that for all $k \in [2 : n-1]$, there exists a constant $M_k(t_2)$ depending only on $L_1(t_2), L_2(t_2)$ and the number of DDIM steps such that $\|G_k(x_1, \theta')\| \leq M_k(t_2)$. First, the constant bounding $\|G_{n-1}(x_1, \theta')\|$ is trivial. Assume then that $\|G_{k+1}(x_1, \theta')\| \leq M_{k+1}(t_2)$. Taking norms and applying the inequality (26) with $t = t_{k+1} \geq t_2$ and $s = t_k$ yields

$$\|G_k(x_1, \theta')\| \leq L_1(t_2) + L_2(t_2)\, \|G_{k+1}(x_1, \theta')\| . \tag{27}$$

and thus $\|G_k(x_1, \theta')\| \leq M_k(t_2) := L_1(t_2) + L_2(t_2)M_{k+1}(t_2)$, which shows the result. □

## A.3. Proof of the main results

*Proof of Proposition 2.* **Step 1: Jacobian bounds on a compact set.** Let $x \notin \mathsf{H}$, and recall the margin function writes $m(x) = \min_{j \neq k^*(x)}(x^{k^*(x)} - x^j)$ with $k^*(x) := \mathrm{argmax}_{j \in [K]} x^j$. By definition of $\mathsf{H}$, it holds then that $m(x) > 0$.

Now consider the logit margin defined for all $j \neq k^*(x)$ by $\Delta_t^j(x, \theta) := (\theta^{k^*(x)} - \theta^j) + c_t(x^{k^*(x)} - x^j)$. Then, letting $B(\theta) := \max_{(i,j) \in [K]^2} |\theta^i - \theta^j|$, we have that

$$\Delta_t^j(x, \theta) \geq -B(\theta) + c_t m(x) .$$

Since $\lim_{t \to 0} c_t = \infty$ by (**A1**), there exists $t_\star(\theta, x)$ such that for all $t < t_\star(\theta, x)$, $\Delta_t^j(x, \theta) \geq c_t m(x)/2$ and thus

$$\min_{j \neq k^*(x)} \Delta_t^j(x, \theta) = m(\theta + c_t x) \geq c_t m(x)/2 ,$$

where we have used that $k^*(\theta + c_t x) = k^*(x)$ since $\Delta_t^j(x, \theta) > 0$ for all $j \neq k^*(x)$. Now define for $t_1 < t_\star(\theta, x)$, $p^{\max}(x, t_1) = \max_{j \in [K]} \hat{x}_0^\theta(x, t_1)^j$ and we recall that $\hat{x}_0^\theta(x, t_1) := \mathrm{softmax}(\theta + c_{t_1} x) \in \Delta^{K-1}$. Applying Lemma 3 with $z = \theta + c_{t_1} x$, we obtain

$$1 - p^{\max}(x, t_1) \leq (K-1) \exp\left(-m(\theta + c_{t_1} x)\right) \leq (K-1) \exp\left(-\frac{m(x)}{2} c_{t_1}\right) .$$

Hence by Lemma 4, for the covariance (16) we have that

$$\left\|\Sigma_{t_1}(x)\right\| \leq 2K\left(1 - p^{\max}(x, t_1)\right) \leq 2K(K-1) \exp\left(-\frac{m(x)}{2} c_{t_1}\right) .$$

Using the gradient identities in Lemma 1 $\mathrm{J}_x T_{0|t_1}^\theta(x) = c_{t_1} \Sigma_{t_1}^\theta(x)$ and $\mathrm{J}_\theta T_{0|t_1}^\theta(x) = \Sigma_{t_1}^\theta(x)$ then for $t_1 \in (0, t_\star(\theta, x))$, we have the following bounds

$$\left\|\mathrm{J}_x T_{0|t_1}^\theta(x)\right\| \leq c_{t_1} M_K \exp(-m(x)c_{t_1}/2) , \tag{28}$$

$$\left\|\mathrm{J}_\theta T_{0|t_1}^\theta(x)\right\| \leq M_K \exp(-m(x)c_{t_1}/2), \tag{29}$$

with $M_K := 2K(K-1)$.

**Step 2: chain rule for the parameter gradient.** For any $x_1 \in \mathbb{R}^K$, $T_0^\theta(x_1) = T_{0|t_1}^\theta\left(T_{t_1}^\theta(x_1)\right)$ and thus for any $\theta' \in \mathbb{R}^K$,

$$\left.\mathrm{J}_\theta T_0^\theta(x_1)\right|_{\theta=\theta'} = \left.\mathrm{J}_\theta T_{0|t_1}^\theta\left(T_{t_1}^{\theta'}(x_1)\right)\right|_{\theta=\theta'} + \mathrm{J}_x T_{0|t_1}^{\theta'}\left(T_{t_1}^{\theta'}(x_1)\right) \cdot \left.\mathrm{J}_\theta T_{t_1}^\theta(x_1)\right|_{\theta=\theta'} .$$

Hence, taking the norms, we get

$$\left\|\left.\mathrm{J}_\theta T_0^\theta(x_1)\right|_{\theta=\theta'}\right\| \leq \left\|\left.\mathrm{J}_\theta T_{0|t_1}^\theta\left(T_{t_1}^{\theta'}(x_1)\right)\right|_{\theta=\theta'}\right\| + \left\|\mathrm{J}_x T_{0|t_1}^{\theta'}\left(T_{t_1}^{\theta'}(x_1)\right)\right\|\left\|\left.\mathrm{J}_\theta T_{t_1}^\theta(x_1)\right|_{\theta=\theta'}\right\|$$

By Lemma 5, there exists a finite constant $M(t_2)$ (depending only on $t_2, K$ and the schedule) such that

$$\sup_{t_1 \in (0,t_2)} \sup_{x_1 \in \mathbb{R}^K} \left\|\left.\mathrm{J}_\theta T_{t_1}^\theta(x_1)\right|_{\theta=\theta'}\right\| \leq M(t_2) .$$

Finally, since by assumptions $x_1 \in \mathbb{R}^K$ is such that $T_{t_1}^\theta(x_1) \notin \mathsf{H}$, we get by applying the bounds (28) and (29)

$$\left\|\left.\mathrm{J}_\theta T_0^\theta(x_1)\right|_{\theta=\theta'}\right\| \leq (1 + c_{t_1} M(t_2)) M_K \exp\left(-m(T_{t_1}^{\theta'}(x_1))c_{t_1}/2\right) .$$

which yields the result. □

*Proof of Proposition 1.* The proof is an immediate consequence of Lemma 2 and Proposition 2. □

# B. Hard STRAIGHT-THROUGH and REINMAX estimators

For completeness, we derive the REINMAX gradient estimator from first principles and recover the simpler expression in (5). We first consider the case $L = 1$. We will then extend to the case $L > 1$. The categorical distribution is parameterized by the vector of logits $\varphi \in \mathbb{R}^K$, with

$$\pi_\theta = \varpi_{\varphi_\theta}, \qquad \text{where} \quad \varpi_\varphi(e_i) := \frac{\exp(\varphi^i)}{\sum_{j=1}^K \exp(\varphi^j)}, \quad i \in [K].$$

Let

$$F(\theta) := \mathbb{E}_{\pi_\theta}[f(X)] = \mathbb{E}_{\varpi_{\varphi_\theta}}[f(X)],$$

where $\varphi_\theta : \mathbb{R}^m \to \mathbb{R}^K$ is differentiable. By the chain rule,

$$\nabla_\theta F(\theta) = \left(\mathrm{J}_\theta \varphi_\theta\right)^\top \nabla_\varphi \mathbb{E}_{\varpi_\varphi}[f(X)]\Big|_{\varphi = \varphi_\theta} \in \mathbb{R}^m. \tag{30}$$

It therefore suffices to consider the parametrization in terms of logits, from which the results for all parameterizations can be readily derived when the logit vectors depend on a parameter $\theta$.

## B.1. Hard STRAIGHT-THROUGH estimator

By definition, we have

$$\mathbb{E}_{\varpi_\varphi}[f(X)] = \sum_{i=1}^K f(e_i)\varpi_\varphi(e_i) \Rightarrow \nabla_\varphi \mathbb{E}_{\varpi_\varphi}[f(X)] = \sum_{i=1}^K f(e_i)\nabla_\varphi \varpi_\varphi(e_i)$$

Using a baseline subtraction, chosen here as $\mathbb{E}_{\varpi_\varphi}[f(X)]$, and exploiting the identities $\sum_{j=1}^K \varpi_\varphi(e_j) = 1$ and $\sum_{j=1}^K \nabla_\varphi \varpi_\varphi(e_j) = 0$, we can rewrite the gradient as

$$\nabla_\varphi \mathbb{E}_{\varpi_\varphi}[f(X)] = \sum_{i,j=1}^K \left(f(e_i) - f(e_j)\right) \nabla_\varphi \varpi_\varphi(e_i) \varpi_\varphi(e_j). \tag{31}$$

This symmetric form is the starting point used by the authors to motivate the first-order approximation interpretation of the straight-through estimator. Indeed, applying a first-order Taylor expansion of $f$ around $e_j$ yields

$$f(e_i) - f(e_j) \approx \nabla_x f(e_j)^\top (e_i - e_j).$$

Substituting this approximation into the symmetric expression gives

$$\nabla_\varphi \mathbb{E}_{\varpi_\varphi}[f(X)] \approx \sum_{i,j=1}^K \nabla_x f(e_j)^\top (e_i - e_j) \nabla_\varphi \varpi_\varphi(e_i) \varpi_\varphi(e_j).$$

It can then be shown that the expectation of the straight-through gradient estimator (4) coincides with the right-hand side, which explains the interpretation of straight-through as an unbiased estimator of a first-order approximation of the true gradient. Define

$$\widehat{\nabla}_\varphi^{\mathrm{ST}} F(X; \varphi) = \mathrm{J}_\varphi \mathbb{E}_{\varpi_\varphi}[X]^\top \nabla_x f(X) = \mathbb{Cov}_{\varpi_\varphi}[X]\nabla_x f(X). \tag{32}$$

**Lemma 6.** *It holds that*

$$\mathbb{E}_{\varpi_\varphi}\left[\widehat{\nabla}_\varphi^{ST} F(X; \varphi)\right] = \mathrm{J}_\varphi \mathbb{E}_{\varpi_\varphi}[X]^\top \mathbb{E}_{\varpi_\varphi}[\nabla_x f(X)] = \sum_{i,j=1}^K \nabla_x f(e_i)^\top (e_j - e_i)\nabla_\varphi \varpi_\varphi(e_j) \varpi_\varphi(e_i).$$

*Proof.* Since $\mathbb{E}_{\varpi_\varphi}[X] = \sum_{j=1}^K e_j \, \varpi_\varphi(e_j)$, its Jacobian with respect to $\varphi$ is $\mathrm{J}_\varphi \, \mathbb{E}_{\varpi_\varphi}[X] = \sum_{j=1}^K e_j \, \nabla_\varphi \varpi_\varphi(e_j)^\top$. Taking expectations in the definition of the straight-through estimator (4) then yields

$$\mathbb{E}_{\varpi_\varphi}\left[\widehat{\nabla}_\varphi^{\mathrm{ST}} F(X;\varphi)\right] = \sum_{i=1}^K \left(\mathrm{J}_\varphi \, \mathbb{E}_{\varpi_\varphi}[X]\right)^\top \nabla_x f(e_i)\, \varpi_\varphi(e_i)$$

$$= \sum_{i,j=1}^K \nabla_\varphi \varpi_\varphi(e_j)\, e_j^\top \nabla_x f(e_i)\, \varpi_\varphi(e_i).$$

Using the identity $\sum_{j=1}^K \nabla_\varphi \varpi_\varphi(e_j) = 0$, which follows from normalization of the categorical distribution, we may perform a baseline subtraction and replace $e_j$ by $(e_j - e_i)$, giving

$$\mathbb{E}_{\varpi_\varphi}\left[\widehat{\nabla}_\varphi^{\mathrm{ST}} F(X;\varphi)\right] = \sum_{i,j=1}^K \nabla_\varphi \varpi_\varphi(e_j)\, (e_j - e_i)^\top \nabla_x f(e_i)\, \varpi_\varphi(e_i)$$

$$= \sum_{i,j=1}^K \nabla_x f(e_i)^\top (e_j - e_i)\, \nabla_\varphi \varpi_\varphi(e_j)\, \varpi_\varphi(e_i),$$

which is the claimed expression. $\qquad\square$

We now extend these results to $L > 1$. In this case $f(x) = f(x^1, \dots, x^L)$ where $x^k \in \mathsf{V}^K$, for $k \in [L]$. We have that, for any $k \in [L]$,
$$\mathbb{E}_{\bigotimes_{\ell\in[L]} \varpi_{\varphi\ell}}[f(X^1, \dots, X^L)] = \mathbb{E}_{\varpi_{\varphi k}}[f^k(X^k)]\,,$$

where we have defined
$$f^k(x^k) = \mathbb{E}_{\bigotimes_{\ell\in[L]\setminus\{\ell\}} \varpi_{\varphi\ell}}[f(X^1, \dots, X^{k-1}, x^k, X^{k+1}, \dots, X^L)]\,. \tag{33}$$

Now define $\varpi_\varphi = \bigotimes_{\ell\in[L]} \varpi_{\varphi\ell}$. Consider the STRAIGHT-THROUGH estimator

$$\widehat{\nabla}_\varphi^{\mathrm{ST}} F(X;\varphi) = \mathrm{J}_\varphi \mathbb{E}_{\varpi_\varphi}[X]^\top \nabla_x f(X). \tag{34}$$

The matrix $\mathrm{J}_\varphi \mathbb{E}_{\varpi_\varphi}[X]^\top$ is a $L \times L$ block-diagonal matrix, whose $k$-th $K \times K$ diagonal block is given by $\mathrm{J}_{\varphi^k} \mathbb{E}_{\varpi_{\varphi k}}[X^k]$. Similarly, the vector $\nabla_x f(x)$ is a $L \times 1$ block vector, whose $k$-th $K \times 1$ block is given by

$$\left[\mathbb{E}_{\varpi_\varphi}[\nabla_x f(X)]\right]^k = \mathbb{E}_{\varpi_{\varphi k}}[\nabla_{x^k} f^k(X^k)].$$

Applying Lemma 6, we get

$$\left[\mathbb{E}_{\varpi_\varphi}[\widehat{\nabla}_\varphi^{\mathrm{ST}} F(X;\varphi)]\right]^k = \sum_{i,j=1}^K \nabla_x f^k(e_i)^\top (e_j - e_i)\nabla_{\varphi^k} \varpi_{\varphi^k}(e_j)\, \varpi_{\varphi^k}(e_i)$$

which is a proxy of

$$\left[\nabla_\varphi \mathbb{E}_{\varpi_\varphi}[f(X)]\right]^k = \nabla_{\varphi^k} \mathbb{E}_{\varpi_{\varphi k}}[f^k(X^k)] = \sum_{i,j=1}^K \left(f^k(e_i) - f^k(e_j)\right) \nabla_{\varphi^k} \varpi_{\varphi^k}(e_i)\, \varpi_{\varphi^k}(e_j)\,.$$

## B.2. REINMAX **estimator**

We again focus first on the case $L = 1$. The extension to general $L$ follows exactly along the same lines than for the hard STRAIGHT-THROUGH estimator. The baisc idea is to consider a second-order approximation of (31) based on Heun's method. Heun's method, also known as the explicit trapezoidal rule, is a second-order Runge–Kutta scheme that improves a first-order approximation by averaging gradients evaluated at the two endpoints of a step. In our discrete setting, this yields a symmetric approximation that averages the gradients at $e_i$ and $e_j$.

Applying this principle yields the second-order approximation

$$\hat{\nabla}_\varphi^{\text{2nd}}\mathbb{E}_{\varpi_\varphi}[f(X)] := \sum_{i,j=1}^K \frac{1}{2}\left(\nabla_x f(e_i) + \nabla_x f(e_j)\right)^\top (e_i - e_j)\, \nabla_\varphi \varpi_\varphi(e_i)\, \varpi_\varphi(e_j)\,. \tag{35}$$

If $f : \mathbb{R}^m \to \mathbb{R}$ be quadratic, then it is easily shown that for all $x, y \in \mathbb{R}^K$,

$$f(y) - f(x) = \frac{1}{2}\left(\nabla_x f(y) + \nabla_x f(x)\right)^\top (y - x).$$

We now derive the REINMAX estimator by rewriting (35) in a more explicit and implementable form. For all $(i, k) \in [K]^2$, we get

$$\partial_{\varphi^k}\, \varpi_\varphi(e_i) = \varpi_\varphi(e_i)\,(\delta_{ik} - \varpi_\varphi(e_k)), \tag{36}$$

where $\delta_{ik}$ denotes the Kronecker symbol.

Define the second-order approximation with respect to $\varphi$ as

$$\hat{\nabla}_\varphi^{\text{2nd}}\mathbb{E}_{\varpi_\varphi}[f(X)] := \sum_{i,j=1}^K \frac{1}{2}\left(\nabla_x f(e_i) + \nabla_x f(e_j)\right)^\top (e_i - e_j)\, \nabla_\varphi \varpi_\varphi(e_i)\, \varpi_\varphi(e_j)\,. \tag{37}$$

Substituting (36) into (37) and extracting the $k$-th coordinate yields

$$\left[\hat{\nabla}_\varphi^{\text{2nd}}\mathbb{E}_{\varpi_\varphi}[f(X)]\right]^k = \frac{1}{2}\sum_{i,j=1}^K \left(\nabla_x f(e_i) + \nabla_x f(e_j)\right)^\top (e_i - e_j)\varpi_\varphi(e_i)(\delta_{ik} - \varpi_\varphi(e_k))\, \varpi_\varphi(e_j). \tag{38}$$

The term proportional to $-\varpi_\varphi(e_k)$ vanishes. Indeed,

$$\sum_{i,j=1}^K \left(\nabla_x f(e_i) + \nabla_x f(e_j)\right)^\top (e_i - e_j)\, \varpi_\varphi(e_i)\, \varpi_\varphi(e_j) = 0,$$

by antisymmetry in $(i, j)$. Retaining only the contribution from $\delta_{ik}$ gives the explicit second-order expression

$$\left[\hat{\nabla}_\varphi^{\text{2nd}}\mathbb{E}_{\varpi_\varphi}[f(X)]\right]^k = \frac{1}{2}\sum_{j=1}^K \left(\nabla_x f(e_k) + \nabla_x f(e_j)\right)^\top (e_k - e_j)\, \varpi_\varphi(e_k)\, \varpi_\varphi(e_j)\,, \tag{39}$$

We now show how this quantity can be rewritten as an expectation involving a single evaluation of $\nabla_x f$. We use the following elementary Lemma:

**Lemma 7.** *For any $k \in [K]$, we get*

$$\left[\hat{\nabla}_\varphi^{\text{2nd}}\mathbb{E}_{\varpi_\varphi}[f(X)]\right]^k = \left[\hat{\nabla}_\varphi^{\text{2nd}}\mathbb{E}_{\varpi_\varphi}[f(X)]\right]^k \quad = \frac{1}{2}\mathbb{E}_{\varpi_\varphi}\left[\nabla_x f(X)^\top \left\{\varpi_\varphi(e_k)(e_k - X) + \langle X, e_k\rangle(X - \mathbb{E}_{\varpi_\varphi}[X])\right\}\right].$$

*Proof.* Write $p_i := \varpi_\varphi(e_i)$ and denote $g_i := \nabla_x f(e_i) \in \mathbb{R}^K$. Since $X$ is categorical on the one-hot vectors, we will repeatedly use the identity

$$\mathbb{E}_{\varpi_\varphi}[\psi(X)] = \sum_{j=1}^K p_j\, \psi(e_j) \quad \text{for any function } \psi \text{ on } \{e_1, \ldots, e_K\}.$$

**Step 1: split the explicit sum into two contributions.** Starting from the explicit expression and expanding the dot product gives

$$\left[\hat{\nabla}_\varphi^{\text{2nd}}\mathbb{E}_{\varpi_\varphi}[f(X)]\right]^k = \frac{1}{2}\sum_{j=1}^K \left(g_k^\top (e_k - e_j) + g_j^\top (e_k - e_j)\right) p_k\, p_j$$

$$= \underbrace{\frac{1}{2}\, p_k\, g_k^\top \sum_{j=1}^K p_j (e_k - e_j)}_{:= T_1} + \underbrace{\frac{1}{2}\, p_k \sum_{j=1}^K p_j\, g_j^\top (e_k - e_j)}_{:= T_2}.$$

**Step 2: rewrite $T_2$ as an expectation.** Using the categorical expectation identity with $\psi(x) = \nabla_x f(x)^\top (e_k - x)$, we obtain

$$\mathbb{E}_{\varpi_\varphi}[\nabla_x f(X)^\top (e_k - X)] = \sum_{j=1}^{K} p_j\, g_j^\top (e_k - e_j).$$

Therefore,

$$T_2 = \frac{1}{2}\, p_k\, \mathbb{E}_{\varpi_\varphi}[\nabla_x f(X)^\top (e_k - X)] = \frac{1}{2}\, \mathbb{E}_{\varpi_\varphi}\Big[ p_k\, \nabla_x f(X)^\top (e_k - X)\Big],$$

since $p_k$ is constant with respect to $X$.

**Step 3: rewrite $T_1$ as an expectation.** First note that

$$\sum_{j=1}^{K} p_j(e_k - e_j) = e_k - \sum_{j=1}^{K} p_j e_j = e_k - \mu.$$

Hence

$$T_1 = \frac{1}{2}\, p_k\, g_k^\top (e_k - \mu).$$

Now use that for one-hot $X$, the scalar $\langle X, e_k \rangle$ is the indicator $\mathbb{1}\{X = e_k\}$. In particular,

$$\langle e_j, e_k \rangle = \delta_{jk}, \qquad \text{and} \qquad \langle X, e_k \rangle\, \nabla_x f(X)^\top (X - \mu) = \begin{cases} g_k^\top (e_k - \mu), & \text{if } X = e_k, \\ 0, & \text{otherwise.} \end{cases}$$

Therefore,

$$\mathbb{E}_{\varpi_\varphi}\Big[ \langle X, e_k \rangle\, \nabla_x f(X)^\top (X - \mu)\Big] = \sum_{j=1}^{K} p_j\, \langle e_j, e_k \rangle\, g_j^\top (e_j - \mu)$$
$$= p_k\, g_k^\top (e_k - \mu),$$

which implies

$$T_1 = \frac{1}{2}\, \mathbb{E}_{\varpi_\varphi}\Big[ \langle X, e_k \rangle\, \nabla_x f(X)^\top (X - \mu)\Big].$$

We conclude by combining the identities for $T_1$ and $T_2$ yields the claimed expectation form. $\qquad\square$

We will now establish an alternative expression for $[\hat{\nabla}_\varphi^{2nd} \mathbb{E}_{\varpi_\varphi}[f(X)]]^k$.

**Lemma 8.** *For any $k \in [K]$, we get*

$$[\hat{\nabla}_\varphi^{2nd} \mathbb{E}_{\varpi_\varphi}[f(X)]]^k$$
$$= \mathbb{E}_{\varpi_\varphi}\left[ \nabla_x f(X)^\top \left\{ 2\frac{\varpi_\varphi(e_k) + \langle X, e_k \rangle}{2}\Big(e_k - \sum_{i=1}^{K} e_i\, \frac{\varpi_\varphi(e_i) + \langle X, e_i \rangle}{2}\Big) - \frac{\varpi_\varphi(k)}{2}\Big(e_k - \sum_{i=1}^{K} e_i\, \varpi_\varphi(e_i)\Big) \right\} \right]. \quad (40)$$

*Proof.* We establish the identity pointwise and then the equality of expectations follows immediately. Let $X \sim \varpi_\varphi$ be categorical on $\{e_1, \ldots, e_K\}$ and denote $p_i := \varpi_\varphi(e_i)$ and $\mu := \mathbb{E}_{\varpi_\varphi}[X] = \sum_{i=1}^{K} p_i e_i$. Since $X$ is one-hot, for each $i \in [K]$ we have $\langle X, e_i \rangle \in \{0, 1\}$ and $X = \sum_{i=1}^{K} e_i \langle X, e_i \rangle$. Define the "averaged" probabilities (used in REINMAX)

$$q_i(X) := \frac{p_i + \langle X, e_i \rangle}{2}, \qquad i \in [K].$$

Then

$$\sum_{i=1}^{K} q_i(X) = \frac{1}{2}\sum_{i=1}^{K} p_i + \frac{1}{2}\sum_{i=1}^{K} \langle X, e_i \rangle = \frac{1}{2}\cdot 1 + \frac{1}{2}\cdot 1 = 1, \qquad (41)$$

so $q(X)$ is a valid probability vector. Moreover,

$$\sum_{i=1}^{K} e_i \, q_i(X) = \frac{1}{2} \sum_{i=1}^{K} p_i \, e_i + \frac{1}{2} \sum_{i=1}^{K} e_i \, \langle X, e_i \rangle = \frac{\mu + X}{2}. \tag{42}$$

For each fixed $k \in [K]$, define

$$A(X) := p_k(e_k - X) + \langle X, e_k \rangle \, (X - \mu)$$

and

$$B(X) := 2 \frac{p_k + \langle X, e_k \rangle}{2} \Big( e_k - \sum_{i=1}^{K} e_i \, \frac{p_i + \langle X, e_i \rangle}{2} \Big) - \frac{p_k}{2} \Big( e_k - \sum_{i=1}^{K} e_i \, p_i \Big).$$

Using (42) and $\sum_{i=1}^{K} e_i \, p_i = \mu$, we rewrite $B(X)$ as

$$
\begin{aligned}
B(X) &= \big( p_k + \langle X, e_k \rangle \big) \Big( e_k - \frac{\mu + X}{2} \Big) - \frac{p_k}{2} \, (e_k - \mu) \\
&= \big( p_k + \langle X, e_k \rangle \big) e_k - \frac{1}{2} \big( p_k + \langle X, e_k \rangle \big) \mu - \frac{1}{2} \big( p_k + \langle X, e_k \rangle \big) X - \frac{p_k}{2} e_k + \frac{p_k}{2} \mu \\
&= \frac{p_k}{2} e_k + \langle X, e_k \rangle \, e_k - \frac{\langle X, e_k \rangle}{2} \mu - \frac{p_k}{2} X - \frac{\langle X, e_k \rangle}{2} X.
\end{aligned}
$$

Now use the one-hot property: if $\langle X, e_k \rangle = 1$, then $X = e_k$, hence $\langle X, e_k \rangle \, e_k = \langle X, e_k \rangle \, X$; if $\langle X, e_k \rangle = 0$, both sides are 0. Therefore, in all cases,

$$\langle X, e_k \rangle \, e_k = \langle X, e_k \rangle \, X. \tag{43}$$

Substituting (43) into the expression for $B(X)$ gives

$$
\begin{aligned}
B(X) &= \frac{p_k}{2} e_k - \frac{p_k}{2} X + \frac{\langle X, e_k \rangle}{2} X - \frac{\langle X, e_k \rangle}{2} \mu \\
&= \frac{1}{2} \Big( p_k(e_k - X) + \langle X, e_k \rangle \, (X - \mu) \Big) = \frac{1}{2} \, A(X).
\end{aligned}
$$

Since $B(X) = \frac{1}{2} A(X)$ holds pointwise, we obtain

$$\frac{1}{2} \, \mathbb{E}_{\varpi_\varphi} \big[ \nabla_x f(X)^\top A(X) \big] = \mathbb{E}_{\varpi_\varphi} \big[ \nabla_x f(X)^\top B(X) \big],$$

which concludes the proof. $\qquad \square$

Recalling that

$$\mathrm{J}_\varphi \, \mathbb{E}_{\varpi_\varphi}[X] = \mathrm{Diag}\big( \mathbb{E}_{\varpi_\varphi}[X] \big) - \mathbb{E}_{\varpi_\varphi}[X] \, \mathbb{E}_{\varpi_\varphi}[X]^\top,$$

and defining the conditional distribution $\varpi_\varphi(e_i|x) := (\varpi_\varphi(e_i) + \langle x, e_i \rangle)/2$, we obtain the standard REINMAX estimator

$$\widehat{\nabla}_\varphi^{\mathrm{RM}} G(X, \varphi) := \Big[ 2 \, \mathbb{Cov}_{\varpi_\varphi(\cdot|X)}(\widetilde{X}) - \frac{1}{2} \, \mathbb{Cov}_{\varpi_\varphi}(X) \Big] \nabla_x f(X) \,, \tag{44}$$

which satisfies

$$\hat{\nabla}_\varphi^{\mathrm{2nd}} \mathbb{E}_{\varpi_\varphi}[f(X)] = \mathbb{E}_{\varpi_\varphi} \big[ \widehat{\nabla}_\varphi^{\mathrm{RM}} G(X; \varphi) \big]. \tag{45}$$

Since Heun's method is exact for quadratic functions, this directly explains why REINMAX is exact for quadratic objectives and can be interpreted as a principled second-order correction of the straight-through estimator.

Lemma 9 shows that

$$\mathbb{Cov}_{\varpi_\varphi(\cdot|x)}(\widetilde{X}) = \frac{1}{2} \mathbb{Cov}_{\varpi_\varphi}(X) + \frac{1}{4} (x - \mathbb{E}_{\varpi_\varphi}[X])(x - \mathbb{E}_{\varpi_\varphi}[X])^\top \,,$$

and plugging in (44) we recover (5); *i.e.*

$$\widehat{\nabla}_\varphi^{\mathrm{RM}} G(X; \varphi) = \frac{1}{2} \big\{ \mathbb{Cov}_{\varpi_\varphi}(X) + (X - \mathbb{E}_{\varpi_\varphi}[X])(X - \mathbb{E}_{\varpi_\varphi}[X])^\top \big\} \nabla_x f(X) \tag{46}$$

**Lemma 9.** *Let $P = \frac{1}{2}P_1 + \frac{1}{2}P_2$ be a mixture distribution on $\mathbb{R}^d$. Denote $\mu_i := \mathbb{E}_{P_i}[X]$ and $\Sigma_i := \mathbb{C}\mathrm{ov}_{P_i}(X)$ for $i \in \{1, 2\}$. Then the mean $\mu$ and covariance $\Sigma$ of $P$ are*

$$\mu = \frac{\mu_1 + \mu_2}{2}, \qquad \Sigma = \frac{\Sigma_1 + \Sigma_2}{2} + \frac{1}{4}(\mu_1 - \mu_2)(\mu_1 - \mu_2)^\top.$$

*Proof.* Let $Z \in \{1, 2\}$ be the component indicator, with $\mathbb{P}(Z = 1) = \mathbb{P}(Z = 2) = \frac{1}{2}$, and let $X \mid (Z = i) \sim P_i$. Then

$$\mathbb{E}[X \mid Z = i] = \mu_i, \qquad \mathbb{C}\mathrm{ov}(X \mid Z = i) = \Sigma_i, \qquad i \in \{1, 2\}.$$

By the law of total expectation,

$$\mu = \mathbb{E}[X] = \mathbb{E}\big[\mathbb{E}[X \mid Z]\big] = \tfrac{1}{2}\mu_1 + \tfrac{1}{2}\mu_2.$$

By the law of total covariance,

$$\Sigma = \mathbb{C}\mathrm{ov}(X) = \mathbb{E}\big[\mathbb{C}\mathrm{ov}(X \mid Z)\big] + \mathbb{C}\mathrm{ov}\big(\mathbb{E}[X \mid Z]\big).$$

The first term is

$$\mathbb{E}\big[\mathbb{C}\mathrm{ov}(X \mid Z)\big] = \tfrac{1}{2}\Sigma_1 + \tfrac{1}{2}\Sigma_2.$$

For the second term, since $\mathbb{E}[X \mid Z] = \mu_Z$ takes values $\mu_1$ and $\mu_2$,

$$\mathbb{C}\mathrm{ov}(\mu_Z) = \mathbb{E}[\mu_Z \mu_Z^\top] - \mu\mu^\top = \tfrac{1}{2}(\mu_1\mu_1^\top + \mu_2\mu_2^\top) - \mu\mu^\top.$$

Using $\mu = \frac{\mu_1 + \mu_2}{2}$, we expand

$$\mu\mu^\top = \frac{1}{4}\big(\mu_1\mu_1^\top + \mu_1\mu_2^\top + \mu_2\mu_1^\top + \mu_2\mu_2^\top\big),$$

hence

$$\tfrac{1}{2}(\mu_1\mu_1^\top + \mu_2\mu_2^\top) - \mu\mu^\top = \frac{1}{4}\big(\mu_1\mu_1^\top - \mu_1\mu_2^\top - \mu_2\mu_1^\top + \mu_2\mu_2^\top\big)$$

$$= \frac{1}{4}(\mu_1 - \mu_2)(\mu_1 - \mu_2)^\top.$$

Combining the two terms gives the stated expression for $\Sigma$. $\qquad\square$

We now consider the extension to $L > 1$. The REINMAX estimator is given by

$$\widehat{\nabla}_\varphi^{\mathrm{RM}}G(X; \varphi) := \frac{1}{2}B_{\varpi_\varphi}(X)\nabla_x f(X), \qquad (47)$$

where $X \sim \varpi_\varphi$ and

$$B_{\varpi_\varphi}(X) = \mathbb{C}\mathrm{ov}_{\varpi_\varphi}(X) + \widehat{C}_\varphi(X)$$

where $\widehat{C}_\varphi(X)$ is block-diagonal with $L$ blocks of size $K \times K$; its $\ell$-th block is $\widehat{C}_\varphi^{(\ell)}(X) := (X^\ell - \mathbb{E}_{\varpi_{\varphi\ell}}[X^\ell])(X^\ell - \mathbb{E}_{\varpi_{\varphi\ell}}[X^\ell])^\top$, $\ell \in [L]$, so that $\mathbb{E}_{\varpi_\varphi}[\widehat{C}_\varphi^{(\ell)}(X)] = \mathbb{C}\mathrm{ov}_{\varpi_\varphi}(X^\ell)$. The matrix $B_{\varpi_\varphi}(X)$ is also block-diagonal with $L$ blocks of size $K \times K$; its $\ell$-th block is

$$B_{\varpi_{\varphi\ell}}^\ell(X^\ell) := [B_{\varpi_\varphi}(X)]^{\ell,\ell} = \mathbb{C}\mathrm{ov}_{\varpi_{\varphi\ell}}(X^\ell) + (X^\ell - \mathbb{E}_{\varpi_{\varphi\ell}}[X^\ell])(X^\ell - \mathbb{E}_{\varpi_{\varphi\ell}}[X^\ell])^\top), \quad \ell \in [L].$$

The vector $\widehat{\nabla}_\varphi^{\mathrm{RM}}G(X; \varphi)$ is a block vector with $L$ blocks of size $K \times 1$ blocks; the $\ell$-th block is given by

$$[\widehat{\nabla}_\varphi^{\mathrm{RM}}G(X; \varphi)]^\ell = \frac{1}{2}B_{\varpi_{\varphi\ell}}^\ell(X^\ell)\nabla_{x^\ell}f(X).$$

Taking the expectation yields to

$$\mathbb{E}_{\varpi_\varphi}\Big[[\widehat{\nabla}_\varphi^{\mathrm{RM}}G(X; \varphi)]^\ell\Big] = \frac{1}{2}\mathbb{E}_{\varpi_\varphi}[B_{\varpi_{\varphi\ell}}^\ell(X^\ell)\nabla_{x^\ell}f^\ell(X^\ell)]$$

where $f^\ell$ is defined in (33). (45) shows that, for $\ell \in [L]$,

$$\hat{\nabla}_\varphi^{\mathrm{2nd}}\Big[\mathbb{E}_{\varpi_\varphi}[f(X)]\Big]^\ell = \hat{\nabla}_\varphi^{\mathrm{2nd}}\mathbb{E}_{\varpi_{\varphi\ell}}[f^\ell(X^\ell)] = \mathbb{E}_{\varpi_\varphi}\Big[[\widehat{\nabla}_\varphi^{\mathrm{RM}}G(X; \varphi)]^\ell\Big]$$

**B.3.** REINDGE, **a** REINMAX **based gradient estimator**

Recall, following the derivations in Section 3.3, that $h_\theta(x_{t_1}) := \sum_{x_0} f(x_0)\, \pi_{0|t_1}^\theta(x_0|x_{t_1})$. The REINDGE gradient estimator at $\theta = \theta'$ is given by

$$\widehat{\nabla}_\theta^{\mathrm{RM}} h_\theta(x_0; T_{t_1}^{\theta'}(X_{t_1}))\big|_{\theta=\theta'} + \mathrm{J}_\theta T_{t_1}^\theta(X_1)^\top\big|_{\theta=\theta'} \widehat{\nabla}_{x_{t_1}}^{\mathrm{ST}} h_\theta(x_0; T_{t_1}^{\theta'}(X_1)) \tag{48}$$

where $X_0 \sim \pi_{0|t_1}^{\theta'}(\cdot|T_{t_1}^{\theta'}(X_{t_1}))$ and following the previous section and (44), we define for any $(x_0, x_{t_1})$,

$$\widehat{\nabla}_{x_{t_1}}^{\mathrm{ST}} h_\theta(x_0; x_{t_1}) := \frac{\alpha_{t_1}}{2\sigma_{t_1}^2}\, \mathbb{C}\mathrm{ov}_{\pi_{0|t_1}^\theta(\cdot|x_{t_1})}(X)\nabla_x f(x_0)$$

$$\widehat{\nabla}_\theta^{\mathrm{RM}} h_\theta(x_0; x_{t_1}) := \frac{1}{2}\, \mathrm{J}_\theta \varphi_\theta^\top B_\theta(x_0; x_{t_1})\nabla_x f(x_0)\ ,$$

and $B_\theta(x_0; x_{t_1}) = \mathbb{C}\mathrm{ov}_{\pi_{0|t_1}^\theta(\cdot|x_{t_1})}(X_0) + \widehat{C}_\theta(x_0; x_{t_1})$, where $\widehat{C}_\theta(x_0; x_{t_1})$ is block-diagonal with $L$ blocks of size $K \times K$; its $\ell$-th block is

$$\widehat{C}_\theta^{(\ell)}(x_0; x_{t_1}) := (x_0^\ell - \mathbb{E}_{\pi_{0|t_1}^\theta(\cdot|x_{t_1})}[X^\ell])(x_0^\ell - \mathbb{E}_{\pi_{0|t_1}^\theta(\cdot|x_{t_1})}[X^\ell])^\top\ .$$

When using a single diffusion step ($t_1 = 1$) and under the boundary condition $\alpha_1 = 0$, the reverse transition no longer depends on the conditioning state (equivalently $X_1$ carries no information about $X_0$), so $h_\theta(x_1)$ is constant in $x_1$ and $h_\theta(x_1) = \mathbb{E}_{\pi_0^\theta}[f(X_0)]$. Hence $\nabla_{x_1} h_\theta(x_1) = 0$ and the second term in (48) vanishes. In that case, REINDGE reduces to precisely the REINMAX gradient estimator for the categorical law $\pi_\theta$. Therefore, in the same way that REDGE recovers STRAIGHT-THROUGH as a special case, REINDGE recovers REINMAX as $t_1 \to 1$.

Since the derivation of this estimator may seem abstract, we provide a code snippet of its implementation in Fig. 11.

## C. DDIM with a general Gaussian reference $\pi_1$

### C.1. Reverse transitions

Let $\pi_0$ be a probability distribution on $\mathbb{R}^d$. Consider the distribution path $(\pi_t)_{t\in[0,1]}$ defined by $\pi_t = \mathrm{Law}(X_t)$, where

$$X_t = \alpha_t X_0 + \sigma_t X_1, \qquad (X_0, X_1) \sim \pi_0 \otimes \pi_1\ , \tag{49}$$

and we take the (more general) reference distribution $\pi_1 = \mathcal{N}(\mu, \Sigma)$ with $\Sigma \in \mathcal{S}_{++}(\mathbb{R}^d)$. Let $(\eta_t)_{t\in[0,1]}$ be a schedule such that $0 \le \eta_t \le \sigma_t$. In the sequel, whenever a distribution is absolutely continuous w.r.t. the Lebesgue measure, we use the same notation for the distribution and its p.d.f.

If $U$ and $V$ are random variables, $U \overset{\mathcal{L}}{=} V$ denotes equality in distribution, *i.e.* $\mathrm{Law}(U) = \mathrm{Law}(V)$. Since $\pi_1$ is Gaussian, we may decompose the Gaussian variable $\sigma_t X_1$ as

$$\sigma_t X_1 \overset{\mathcal{L}}{=} (\sigma_t^2 - \eta_t^2)^{1/2} X_1 + \left(\sigma_t - (\sigma_t^2 - \eta_t^2)^{1/2}\right)\mu + \eta_t \Sigma^{1/2} Z, \qquad (X_1, Z) \sim \pi_1 \otimes \mathcal{N}(0, \mathbf{I}_d)\ . \tag{50}$$

Indeed, both sides have mean $\sigma_t \mu$ and covariance $\sigma_t^2 \Sigma$.

Combining (49) and (50), we obtain $X_t \overset{\mathcal{L}}{=} X_t^\eta$, where we define

$$X_t^\eta = \alpha_t X_0 + (\sigma_t^2 - \eta_t^2)^{1/2} X_1 + \left(\sigma_t - (\sigma_t^2 - \eta_t^2)^{1/2}\right)\mu + \eta_t \Sigma^{1/2} Z_t, \tag{51}$$

$$(X_0, X_1, Z_t) \sim \pi_0 \otimes \pi_1 \otimes \mathcal{N}(0, \mathbf{I}_d)\ .$$

Denote by $q_{t|0,1}^\eta(\cdot|x_0, x_1)$ the conditional distribution of $X_t^\eta$ given $(X_0, X_1) = (x_0, x_1)$. Then clearly from (49) and (51) we have for all $\eta_t \in [0, \sigma_t]$,

$$\pi_t(\mathrm{d}x_t) = \int q_{t|0,1}^\eta(\mathrm{d}x_t|x_0, x_1)\, \pi_0(\mathrm{d}x_0)\pi_1(\mathrm{d}x_1)\ . \tag{52}$$

Now, for $0 \le s < t \le 1$, define the reverse transition

$$\pi_{s|t}^\eta(\mathrm{d}x_s|x_t) := \int q_{s|0,1}^\eta(\mathrm{d}x_s|x_0, x_1)\, \pi_{0,1|t}(\mathrm{d}(x_0, x_1)|x_t) \tag{53}$$

where $\pi_{0,1|t}(\cdot|x_t)$ denotes the conditional distribution of $(X_0, X_1)$ given $X_t = x_t$ under the joint distribution induced by (49). This conditional can be written as $\pi_{0,1|t}(\mathrm{d}(x_0, x_1)|x_t) = \delta_{(x_t - \alpha_t x_0)/\sigma_t}(\mathrm{d}x_1)\,\pi_{0|t}(\mathrm{d}x_0|x_t)$,

$$\pi_{0|t}(\mathrm{d}x_0|x_t) = \frac{\pi_0(\mathrm{d}x_0)\,\mathrm{N}(x_t; \alpha_t x_0 + \sigma_t \mu, \sigma_t^2 \Sigma)}{\pi_t(x_t)}. \tag{54}$$

Indeed, for any bounded measurable function $f$, we get

$$\int f(x_0, x_t, x_1)\,\pi_{0,1|t}(\mathrm{d}(x_0,x_1)|x_t)\pi_t(\mathrm{d}x_t) = \int f(x_0, x_t, x_1)\,\delta_{\frac{x_t - \alpha_t x_0}{\sigma_t}}(\mathrm{d}x_1)\pi_0(\mathrm{d}x_0)\mathrm{N}(x_t; \alpha_t x_0 + \sigma_t\mu, \sigma_t^2\Sigma)\mathrm{d}x_t$$

$$= \int f(x_0, x_t, \frac{x_t - \alpha_t x_0}{\sigma_t})\,\pi_0(\mathrm{d}x_0)\mathrm{N}(x_t; \alpha_t x_0 + \sigma_t\mu, \sigma_t^2\Sigma)\mathrm{d}x_t$$

$$= \int f(x_0, \alpha_t x_0 + \sigma_t\mu + \sigma_t\Sigma^{1/2}z, \mu + \Sigma^{1/2}z)\,\pi_0(\mathrm{d}x_0)\mathrm{N}(z; 0, \mathbf{I}_d)\mathrm{d}z$$

$$= \int f(x_0, \alpha_t x_0 + \sigma_t x_1, x_1)\,\pi_0(\mathrm{d}x_0)\mathrm{N}(x_1; \mu, \Sigma)\mathrm{d}x_1$$

$$= \int f(x_0, x_t, x_1)\,\delta_{\alpha_t x_0 + \sigma_t x_1}(\mathrm{d}x_t)\pi_0(\mathrm{d}x_0)\mathrm{N}(x_1; \mu, \Sigma)\mathrm{d}x_1$$

which shows that

$$\pi_{0,1|t}(\mathrm{d}(x_0, x_1)|x_t)\pi_t(\mathrm{d}x_t) = \delta_{\alpha_t x_0 + \sigma_t x_1}(\mathrm{d}x_t)\pi_0(\mathrm{d}x_0)\pi_1(\mathrm{d}x_1)\,, \tag{55}$$

where the r.h.s. is the joint distribution of the random variables $(X_0, X_t, X_1)$ defined by (49). It then follows that

$$\int \pi_{s|t}^\eta(\mathrm{d}x_s|x_t)\,\pi_t(\mathrm{d}x_t) = \int q_{s|0,1}^\eta(\mathrm{d}x_s|x_0, x_1)\,\pi_{0,1|t}(\mathrm{d}(x_0, x_1)|x_t)\pi_t(\mathrm{d}x_t)$$

$$= \int q_{s|0,1}^\eta(\mathrm{d}x_s|x_0, x_1)\,\pi_0(\mathrm{d}x_0)\pi_1(\mathrm{d}x_1)$$

$$= \pi_s(\mathrm{d}x_s)$$

where the second line follows from integrating the r.h.s. in (55) w.r.t. $x_t$ and the third one from (52). Finally, by noting that

$$\pi_{s|t}^\eta(\mathrm{d}x_s|x_t) = \int q_{s|0,1}^\eta(\mathrm{d}x_s|x_0, x_1)\,\pi_{0,1|t}(\mathrm{d}(x_0, x_1)|x_t)$$

$$= \int \underbrace{q_{s|0,1}^\eta(\mathrm{d}x_s|x_0, \frac{x_t - \alpha_t x_0}{\sigma_t})}_{q_{s|0,t}^\eta(\cdot|x_0, x_t)}\,\pi_{0|t}(\mathrm{d}x_0|x_t)$$

where the defined $q_{s|0,t}^\eta(\cdot|x_0, x_t)$, up to the notation, is exactly the DDIM bridge transition Song et al. (2021, Equation 7) when $\mu = 0_d$ and $\Sigma = \mathbf{I}_d$. Finally, the Gaussian approximation $q_{s|0,t}^\eta(\cdot|\hat{x}_0(x_t, t), x_t)$ used at inference, with $\hat{x}_0(x_t, t) := \int x_0\,\pi_{0|t}(x_0|x_t)\mathrm{d}x_0$, is the one solving

$$\operatorname*{argmin}_{r_{s|t}(\cdot|x_t) \in \mathcal{G}_{\eta_s^2\Sigma}} \mathrm{KL}(\pi_{s|t}^\eta(\cdot|x_t)\|r_{s|t}(\cdot|x_t))\,,$$

where $\mathcal{G}_{\eta_s^2\Sigma} := \{\mathcal{N}(\mu, \eta_s^2\Sigma) :\ \mu \in \mathbb{R}^d\}$ is the set of Gaussian distributions with covariance set to $\eta_s^2\Sigma$.

## C.2. Explicit denoiser for categorical distributions

In this section we extend the derivation in (3.2) to the case where

$$\pi_1 = \bigotimes_{i=1}^L \mathcal{N}(\mu^i, \mathrm{Diag}(v^i)), \qquad \mu^i, v^i \in \mathbb{R}^K, \quad v^{ij} > 0\,.$$

Following (51) and the factorization (2), we still have $\pi_{0|t}^\theta(x_0|x_t) \propto \prod_{i=1}^L \pi_{0|t}^{\theta,i}(x_0^i|x_t^i)$

$$\pi_{0|t}^{\theta,i}(x_0^i|x_t^i) \propto \pi_\theta^i(x_0^i) \, \mathrm{N}\big(x_t^i; \alpha_t x_0^i + \sigma_t \mu^i, \sigma_t^2 \mathrm{Diag}(v^i)\big) . \tag{56}$$

With this structure, the denoiser $\hat{x}_0^\theta(x_t, t) := \sum_{x_0} x_0 \, \pi_{0|t}^\theta(x_0|x_t)$ simplifies to a matrix of posterior probabilities due to the one-hot structure; *i.e.* for any $i \in [L]$ and $j \in [K]$, $\hat{x}_0^\theta(x_t, t)^{ij} = \pi_{0|t}^{\theta,i}(e_j|x_t)$. Using that

$$\mathrm{N}(x_t^i; \alpha_t e_j + \sigma_t \mu^i, \sigma_t^2 \mathrm{Diag}(v^i)) \propto \exp\left( -\frac{1}{2\sigma_t^2} \sum_{k=1}^K \frac{(x_t^{ik} - \alpha_t e_j^k - \sigma_t \mu^{ik})^2}{v^{ik}} \right) ,$$

we expand the quadratic term and drop all terms independent of $j$ to obtain the logits

$$\log \hat{x}_0^\theta(x_t, t)^{ij} = \log \varphi_\theta^{ij} + \frac{\alpha_t}{\sigma_t^2} \frac{x_t^{ij} - \sigma_t \mu^{ij}}{v^{ij}} - \frac{\alpha_t^2}{2\sigma_t^2} \frac{1}{v^{ij}} + C(i,t) . \tag{57}$$

Equivalently, for each $(i,j) \in [L] \times [K]$,

$$\hat{x}_0^\theta(x_t, t)^{ij} = \frac{\pi_\theta^i(e_j) \exp\left( \frac{\alpha_t}{\sigma_t^2 v^{ij}}(x_t - \sigma_t \mu^{ij} - \frac{\alpha_t}{2}) \right)}{\sum_{k=1}^K \pi_\theta^i(e_k) \exp\left( \frac{\alpha_t}{\sigma_t^2 v^{ik}}(x_t - \sigma_t \mu^{ik} - \frac{\alpha_t}{2}) \right)} \tag{58}$$

which yields

$$\hat{x}_0^\theta(x_t, t) = \mathrm{softmax}\big(\varphi_\theta + \frac{\alpha_t \lambda}{\sigma_t^2} \odot (x_t - \sigma_t \mu - \frac{\alpha_t}{2}\mathbf{1})\big). \tag{59}$$

where $\lambda \in \mathbb{R}^{L \times K}$ with $\lambda^{i,j} = 1/v^{i,j}$ and $\mathbf{1} \in \mathbb{R}^{L \times K}$ is the all-ones matrix.

# D. Variational guidance in masked diffusion models

**Masked diffusion models.** We recall the reader that the state space we consider is $\mathsf{X} = \mathsf{V}^L$ where the vocabulary of size $K$, $\mathsf{V}$, is made of $K$ one-hot encoding $e_1, \ldots, e_K$. For all the masked diffusion experiments we assume that the last state $e_K$ in $\mathsf{V}$ is associated to the mask $\mathrm{m}$. We further denote by $m \in \mathsf{X}$ the matrix with all masks, *i.e.* $m^i = \mathrm{m}$. In order to align with the notation from previous works (Sahoo et al., 2024; Shi et al., 2024), we define, for a row-stochastic matrix $\pi$, $\mathrm{Cat}(x; \pi) := \prod_{i=1}^L \langle x^i, \pi^i \rangle$ .

Let $p$ be a target data distribution on $\mathsf{X}$. We further assume that $p(m) = 0$. Similarly to Gaussian diffusion (see Section 3.1), MDMs define a generative procedure for $p$ by specifying a continuous family of marginals $(p_t)_{t \in [0,1]}$ that connects $p$ to the simple reference $\delta_m$. More precisely, the marginals are defined as

$$p_t^\mathsf{d}(x_t) = \sum_{x_0 \in \mathsf{X}} q_{t|0}^\mathsf{d}(x_t|x_0) p(x_0) , \quad \text{where } q_{t|0}^\mathsf{d}(x_t|x_0) := \mathrm{Cat}(x_t; \beta_t x_0 + (1 - \beta_t)m) , \tag{60}$$

where $(\beta_t)_{t \in [0,1]}$ is a decreasing schedule satisfying $\beta_0 = 1$ and $\beta_1 \approx 0$. Define now the reverse transition

$$p_{s|t}^\mathsf{d}(x_s|x_t) := \sum_{x_0 \in \mathsf{X}} q_{s|0,t}^\mathsf{d}(x_s|x_0, x_t) p_{0|t}^\mathsf{d}(x_0|x_t) , \quad \text{where } q_{s|0,t}^\mathsf{d}(x_s|x_0, x_t) := \mathrm{Cat}(x_s; \frac{\beta_s - \beta_t}{1 - \beta_t} x_0 + \frac{1 - \beta_s}{1 - \beta_t} x_t) , \tag{61}$$

and $p_{0|t}^\mathsf{d}(x_0|x_t) := p(x_0) q_{t|0}^\mathsf{d}(x_t|x_0)/p_t^\mathsf{d}(x_t)$ and is a probability distribution following the previous definitions. Next, we

have that

$$
\begin{aligned}
\sum_{x_t} p_{s|t}^{\mathsf{d}}(x_s|x_t)p_t^{\mathsf{d}}(x_t) &= \sum_{x_0,x_t} q_{s|0,t}^{\mathsf{d}}(x_s|x_0,x_t)p_{0|t}^{\mathsf{d}}(x_0|x_t)p_t^{\mathsf{d}}(x_t) \\
&= \sum_{x_0,x_t} q_{s|0,t}^{\mathsf{d}}(x_s|x_0,x_t)p_0(x_0)q_{t|0}^{\mathsf{d}}(x_t|x_0) \\
&= \sum_{x_0} \prod_{i=1}^{L} \sum_{x_t^i} \langle \frac{\beta_s-\beta_t}{1-\beta_t}x_0^i + \frac{1-\beta_s}{1-\beta_t}x_t^i \rangle \langle x_t^i, \beta_t x_0^i + (1-\beta_t)\mathrm{m} \rangle \, p(x_0) \\
&= \sum_{x_0} \prod_{i=1}^{L} \langle \frac{\beta_s-\beta_t}{1-\beta_t}x_0^i + \frac{1-\beta_s}{1-\beta_t}\left(\beta_t x_0^i + (1-\beta_t)\mathrm{m}\right) \rangle \, p(x_0) \\
&= \sum_{x_0} \langle x_s, \beta_s x_0 + (1-\beta_s)m \rangle \, p(x_0) \\
&= p_s^{\mathsf{d}}(x_s)
\end{aligned}
$$

where the third equality follows by linearity of the scalar product and the last one by the definition of the marginal $p_s^{\mathsf{d}}$. As a result, given a sample $X_t \sim p_t^{\mathsf{d}}$, sampling from $p_{s|t}^{\mathsf{d}}(\cdot|X_t)$ yields an exact sample from $p_s^{\mathsf{d}}$. However, sampling from this reverse transition is infeasible in practice because the posterior $p_{0|t}^{\mathsf{d}}(\cdot \mid X_t)$ is intractable. MDMs sidestep this issue by learning a factorized approximation to this posterior via cross-entropy minimization. Concretely, we introduce a factorized posterior that matches the one-dimensional marginals of the target posterior:

$$
\hat{p}_{0|t}^{\mathsf{d}}(\cdot|x_t) \coloneqq \prod_{i=1}^{L} p_{0|t}^{\mathsf{d},i}(x_0^i|x_t) , \quad \text{where} \quad p_{0|t}^{\mathsf{d},i}(x_0^i|x_t) \coloneqq \sum_{x_0^{1:L\setminus i}} p_{0|t}^{\mathsf{d}}(x_0^{1:L}|x_t) . \tag{62}
$$

It is straightforward to show that for all $i \in [L]$, if $x_t^i = \mathrm{m}$ then $p_{0|t}^{\mathsf{d},i}(\cdot|x_t) = \delta_{x_t^i}$. Following (Sahoo et al., 2024) we call this property *carry-over unmasking*.

The factorization (62) can be learned straightforwardly with a neural network, and samples from the resulting approximation can be generated efficiently. We denote the approximation $p_{0|t}^{\mathsf{d},\psi}(\cdot|x_t)$. Plugging this approximation in (61) yields a tractable reverse transition, which, after discretization and iterating it to propagate from one timestep to the next, defines an approximate sampler for the data distribution starting from $\delta_m$.

**Inference-time guidance.** We essentially follow the methodology proposed in (Murata et al., 2025). Let us now assume that the target distribution is $\mu(x_0) \propto \exp(-r(x_0))p(x_0)$, with $r$ a positive reward function, and assume also that we have access to a pre-trained MDM for $p$. In order to have an MDM for $\mu$ we need, following the previous section, to have a factorized approximation of the posterior

$$
\mu_{0|t}^{\mathsf{d}}(x_0|x_t) \propto \mu(x_0)q_{t|0}^{\mathsf{d}}(x_t|x_0) \propto \exp(-r(x_0))p_{0|t}^{\mathsf{d}}(x_0|x_t) .
$$

(Murata et al., 2025) proposes to learn a factorized variational approximation of this posterior by minimizing, at inference time and for each given $x_t$, an objective over the parameters of the factorized family. Crucially, this optimization is performed on the fly for the current test instance and each time step while keeping the pre-trained MDM for $p$ fixed: no additional offline training or dataset-level fine-tuning is required, and the variational parameters are discarded once the sample is produced.

More formally, let $\pi_\theta$ be a factorized distribution (2) parameterized through its logits $\varphi_\theta$. Then at each timestep $t$, (Murata et al., 2025) propose to optimize the KL divergence $\mathrm{KL}(\pi_\theta \| \mu_{0|t}^{\mathsf{d}}(\cdot|x_t))$, which is equivalent to the objective (14) upon replacing $p_{0|t}^{\mathsf{d}}(\cdot|x_t)$ with the pre-trained factorized transition $p_{0|t}^{\mathsf{d},\psi}(\cdot|x_t)$. Once the factorized distribution is optimized, a sample $\hat{X}_0$ is drawn from it and the next sample at timestep $s$ is obtained by sampling $q_{s|0,t}(\cdot|\hat{X}_0, x_t)$ defined previously. Another alternative presented in (Murata et al., 2025) consists in simply sampling the next state from $q_{s|0}(\cdot|\hat{X}_0)$. See Algorithm 2 where we summarize the algorithm proposed in (Murata et al., 2025).

**Algorithm 2** G2D2

**Require:** Pre-trained factorized posterior $p_{0|t}^{\mathsf{d},\psi}$; reward $r$; grid $(\ell_k)_{k=0}^{M-1}$ with $\ell_0 = 0$ and $\ell_{M-1} = 1$; schedule $(\beta_{\ell_k})_{k=0}^{M-1}$; inner optimization steps $J$
1: $x \leftarrow m$
2: **for** $k = M - 1$ **down to** 0 **do**
3:     $\varphi_\theta \leftarrow b$   where $b^{ij} = \log p_{0|\ell_{k+1}}^{\mathsf{d},\psi,i}(e_j|x)$
4:     **for** $j = 1, 2, \ldots, J$ **do**
5:         $\varphi_\theta \leftarrow \text{OPTIMIZE}\Big(\theta; \; \theta \mapsto \mathbb{E}_{\pi_\theta}[r(X_0)] + \text{KL}(\pi_\theta \| p_{0|\ell_{k+1}}^{\mathsf{d},\psi}(\cdot|x))\Big)$
6:     **end for**
7:     Sample $\hat{x}_0 \sim \pi_\theta$
8:     Sample $x_{\ell_k} \sim q_{\ell_k|0}^{\mathsf{d}}(\cdot|\hat{x}_0)$
9:     Set $x \leftarrow x_{\ell_k}$
10: **end for**
11: **return** $x$

# E. Impact of $t_1$ and $n$ on our gradient estimator

In this section, we investigate the impact of the two main hyperparameters of our estimator $n$ and $t_1$ and draw practical conclusion as to how to set them.

### E.1. Impact of $t_1$ and $n$ on the quality of the approximation of $\pi_\theta$

In most experiments we use a small number of time steps, we typically set $n \in \{3, \ldots, 9\}$. This raises the question of whether such coarse discretizations suffice to obtain accurate samples from $\pi_\theta$.

In our setting, the denoiser (and thus the final reverse transition from $t_1$ to 0) is available in closed form and the target distribution is simple. Empirically, this makes a small number of steps sufficient to obtain samples whose empirical law is nearly indistinguishable from $\pi_\theta$.

**Empirical protocol.** Since the categorical distributions we are interested in factorize over the dimensions, it is sufficient to study the case $L = 1$. Therefore, we draw random logits $\theta \in \mathbb{R}^K$ with $K = 100$ and form the target categorical distribution $\pi_\theta = \text{softmax}(\theta)$. For each of 10 seeds, we generate a reference set of 10,000 i.i.d. samples from $\pi_\theta$ and compare it to (i) 10,000 samples produced by REDGE, and (ii) 10,000 samples produced by REDGE-COV. We also generate a second i.i.d. set of 10,000 samples from $\pi_\theta$ to quantify finite-sample fluctuations. For each method and each $(t_1, n)$, we compute the empirical Wasserstein distance between the corresponding empirical laws and the reference empirical law. Results are reported in Fig. 7.

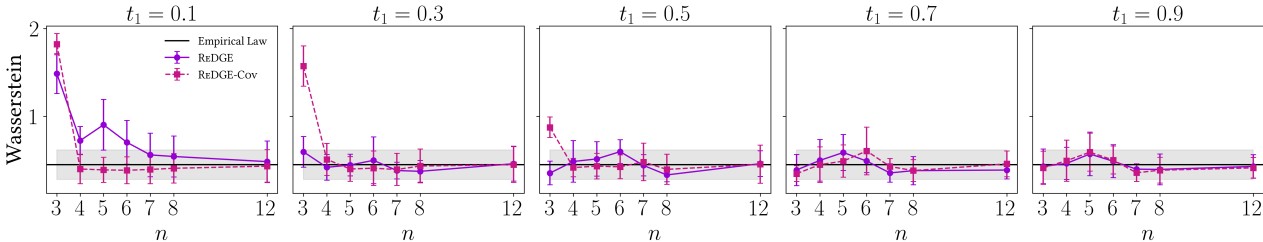

*Figure 7.* Empirical Wasserstein distance between the laws induced by REDGE/REDGE-COV and a reference empirical law drawn from $\pi_\theta$, for different $(t_1, n)$. The gray band shows the mean $\pm$ std. of the Wasserstein distance between two independent empirical draws from $\pi_\theta$ (finite-sample baseline).

**Discussion.** The gray band in Fig. 7 corresponds to the empirical standard deviation between two independent empirical measures drawn from $\pi_\theta$, that was computed using the different sets of i.i.d. samples from $\pi_\theta$. Values within this band indicate that the sampler's empirical law is statistically indistinguishable from the target at this sample size. We observe that for most configurations, both REDGE and REDGE-COV fall within this baseline. The main failure mode occurs for small

$t_1$ combined with very few steps, roughly $t_1 < 0.3$ and $n < 4$.

This dependence on $t_1$ is consistent with how sampling is performed. In the forward pass we first approximate the marginal $X_{t_1}$ by discretizing the diffusion on $[1, t_1]$ with $n$ steps, and then sample $X_0 \sim \pi_{0|t_1}^\theta(\cdot | X_{t_1})$. The latter step is exact in our setting (closed-form denoiser), so the sampling accuracy is governed solely by the discretization error in the approximation of the marginal law of $X_{t_1}$. Holding $n$ fixed, this error increases as the interval length $|1 - t_1|$ increases. Hence the most challenging regime is precisely small $t_1$ with small $n$.

**Remark 2.** *In Fig. 7, we omit* REINDGE *because its sampling procedure is identical to* REDGE*It differs only in the gradient estimator and therefore does not affect the forward-pass approximation quality of $\pi_\theta$.*

Overall, these results suggest that when accurate forward samples from (approximately) $\pi_\theta$ are required, a conservative choice such as $t_1 \geq 0.3$ and $n \geq 4$ is sufficient in this regime. This choice is also aligned with Proposition 1: overly small $t_1$ may lead to vanishing or unstable gradients, even when sampling remains accurate.

Overlooking the mismatch between the sampling distribution induced by a relaxation and the target law $\pi_\theta$ can be benign in some settings, but detrimental in others. A particularly important case is ELBO optimization, where this issue has been discussed in Maddison et al. (2017); Tucker et al. (2017). Indeed, if one naively evaluates the objective on samples produced by a non-exact sampler—such as the soft GUMBEL-SOFTMAX relaxation or our diffusion-based samplers—the resulting Monte Carlo estimates generally do *not* correspond to the ELBO associated with $\pi_\theta$. The reason is that the ELBO contains a KL term defined under the variational distribution, whereas the samples used to estimate the expectation are drawn from a different (relaxed) law. If not accounted for, this discrepancy can decouple apparent optimization progress from actual improvements in the underlying model. We illustrate this phenomenon empirically in the categorical VAE experiments on binarized MNIST (Kingma & Welling; Rezende & Mohamed, 2015).

**Categorical VAE.** Following the setups of Tucker et al. (2017); Grathwohl et al. (2018); Liu et al. (2023a). The encoder maps an input $y \in \{0, 1\}^{784}$ to logits $\varphi_\theta(y) \in \mathbb{R}^{L \times K}$ and defines the mean-field posterior $\pi_\theta(\cdot | y)$ in (2). The decoder maps $z \in \mathbb{R}^{L \times K}$ to pixel logits $\eta_\phi(z) \in \mathbb{R}^{784}$ and defines $p_\phi(\cdot \mid z) = \prod_{j=1}^{784} \text{Bernoulli}(\sigma(\eta_\phi(z)^j))$, where $\sigma$ is the sigmoid function. Given a dataset $\{Y_n\}_{n=1}^N$, we jointly optimize $(\theta, \phi)$ by minimizing the negated ELBO

$$F(\theta; \phi) := -\frac{1}{N} \sum_{n=1}^N \mathbb{E}_{\pi_\theta(\cdot | Y_n)} \big[ \log p_\phi(Y_n \mid Z_n) \big] + \frac{1}{N} \sum_{n=1}^N \text{KL}\big(\pi_\theta(\cdot | X_n) \,\big\|\, p_z\big) \,,$$

where $p_z := \text{Uniform}(\mathsf{X})$ is the discrete uniform prior on $\mathsf{X}$.

This highlights a subtle but important issue. If we use samples from our sampler, we would actually be optimizing $-\frac{1}{N} \sum_{n=1}^N \mathbb{E}_{\hat{\pi}_\theta(\cdot | Y_n)} \big[ \log p_\phi(Y_n \mid Z_n) \big] + \frac{1}{N} \sum_{n=1}^N \text{KL}\big(\pi_\theta(\cdot | Y_n) \,\big\|\, p_z\big)$ where $\hat{\pi}_\theta(\cdot \mid Y_n)$ is the law of $T_0^\theta(X_1)$ which is not exactly equal to $\pi_\theta(\cdot | Y_n)$ unless we are using an infinite number of steps. Therefore the objective that we are optimizing is not an ELBO. We emphasize that this is not specific to our estimator but is an intrinsic fact about any soft-reparameterization of a categorical distribution. Indeed, as discussed in (Maddison et al., 2017; Tucker et al., 2017) this mismatch also happens when using a GUMBEL-SOFTMAX relaxation in its soft version.

To empirically validate this discussion, we report (i) the best training loss computed using samples from each sampler, and (ii) the corresponding "true" loss, computed at each epoch using independent samples drawn directly from $\pi_\theta(\cdot | Y_n)$. Results are shown in Tables 1, 2, 3, and 4. We use a learning rate of $10^{-4}$ and 200 epochs with $N = 200$. They match the above analysis: for small $t_1$ and $n$, the reported training loss can appear artificially improved without a commensurate improvement in the true objective. In contrast, for moderately larger $t_1$ or $n$, the training loss becomes well aligned with the true loss, and our samplers achieve performance comparable to, or better than, the baselines.

*Table 1.* VAE configs filetered by best training loss value (L=24, K=2).

| Sampler | Hyperparameter | Best loss | Best true loss |
|---|---|---|---|
| REDGE | $n = 3, t_1 = 0.1$ | 87.3197 | 174.5299 |
| REINDGE | $n = 3, t_1 = 0.1$ | **87.2011** | 173.2382 |
| REDGE-COV | $n = 3, t_1 = 0.3$ | 87.5007 | 149.6025 |
| GUMBEL-SOFTMAX | $\tau = 0.4$ | 94.4864 | **94.4623** |
| REINMAX | — | 95.5282 | 95.5106 |
| ST | — | 108.1155 | 108.0957 |

*Table 2.* VAE configs filtered by best true loss value (L=24, K=2).

| Sampler | Hyperparameter | Best loss | Best true loss |
|---|---|---|---|
| REDGE | $n = 5, t_1 = 0.3$ | 94.3861 | 94.8035 |
| REINDGE | $n = 3, t_1 = 0.3$ | **92.6446** | **93.6885** |
| REDGE-COV | $n = 7, t_1 = 0.6$ | 93.4340 | 93.8271 |
| GUMBEL-SOFTMAX | $\tau = 0.4$ | 94.4864 | 94.4623 |
| REINMAX | — | 95.5282 | 95.5106 |
| ST | — | 108.1155 | 108.0957 |

*Table 3.* VAE configs filtered by best training loss value (L=48, K=2).

| Sampler | Hyperparameter | Best loss | Best true loss |
|---|---|---|---|
| REDGE | $n = 3, t_1 = 0.1$ | 70.8737 | 171.3284 |
| REINDGE | $n = 3, t_1 = 0.1$ | **71.0220** | 172.2232 |
| REDGE-COV | $n = 3, t_1 = 0.2$ | 71.6501 | 168.2566 |
| GUMBEL-SOFTMAX | $\tau = 0.5$ | 88.1752 | 88.2038 |
| REINMAX | — | 87.7028 | **87.6808** |
| ST | — | 99.1930 | 99.2280 |

*Table 4.* VAE configs filtered by best true loss value (L=48, K=2).

| Sampler | Hyperparameter | Best loss | Best true loss |
|---|---|---|---|
| REDGE | $n = 5, t_1 = 0.4$ | 87.9557 | 88.8152 |
| REINDGE | $n = 5, t_1 = 0.5$ | **86.8424** | **86.9511** |
| REDGE-COV | $n = 5, t_1 = 0.6$ | 87.8160 | 89.0508 |
| GUMBEL-SOFTMAX | $\tau = 0.5$ | 88.1752 | 88.2038 |
| REINMAX | — | 87.7028 | 87.6808 |
| ST | — | 99.1930 | 99.2280 |

## E.2. Impact of $t_1$ and $n$ on the stability and quality of the estimated gradient

We study how the cutoff time $t_1$ and the number of discretization steps $n$ (and their interaction) affect the stability and quality of our gradient estimators.

Empirically, Fig. 4 shows a clear trend: for sufficiently large $t_1$, increasing $n$ has little effect on performance, whereas for $t_1 \to 0$ larger $n$ can *degrade* performance. At first glance this may appear counter-intuitive—in diffusion models, finer discretizations are often beneficial. In our setting, however, the performance drop is explained by gradient instabilities rather than sampling error.

A natural hypothesis is that increasing $n$ forces backpropagation through a longer diffusion trajectory, akin to differentiating through a deeper computational graph, which may lead to vanishing/exploding gradients as it is the case in recurrent neural

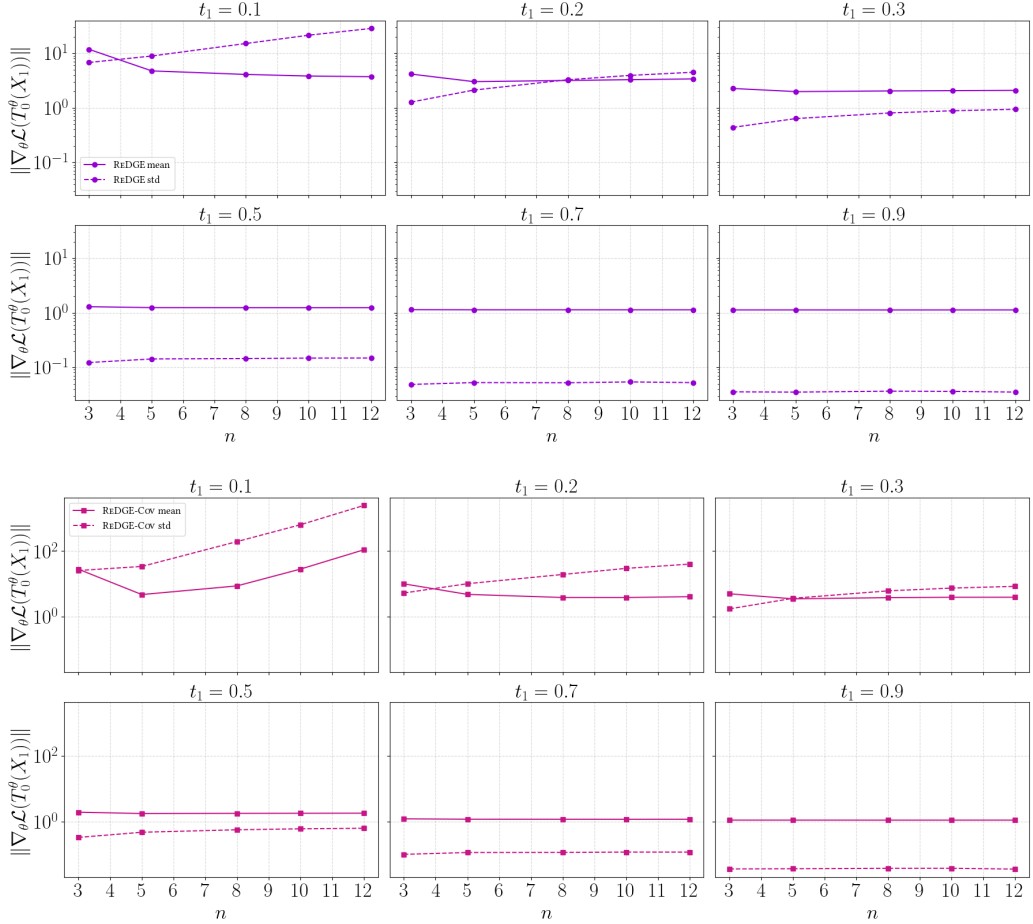

*Figure 8.* Mean and standard deviation (across 500 samples) of $\nabla_\theta \mathcal{L}(X_i)$ as a function of $(t_1, n)$. Top: REDGE, bottom: REDGE-COV.

network training. While plausible, this explanation is incomplete: it does not account for why the effect is pronounced when $t_1$ is small yet largely negligible for $t_1 \gtrsim 0.5$.

Instead, the behavior is consistent with the mechanism behind the proof of Proposition 1. When $t_1$ is close to 0, the reverse transition becomes highly concentrated and gradients associated with the final step (from $t_1$ to 0) can be unstable. Increasing $n$ refines the discretization on $[1, t_1]$, making consecutive times $t_2, t_3, \ldots$ closer to $t_1$. This effectively composes several near-terminal transitions, so that the instabilities are propagated and amplified backward through additional steps. As a result, for small $t_1$, larger $n$ can exacerbate vanishing/exploding behavior and lead to poorer optimization. In contrast, when $t_1$ is bounded away from 0, the law of $X_{t_1}$ is smoother, and the gradient becomes largely insensitive to $n$; in that regime, taking more steps is at best marginally beneficial, and often unnecessary since $n = 3$ already yields near-perfect samples (cf. §E.1) as indicated by Fig. 4.

**Empirical verification.** To corroborate this prediction, we consider the polynomial loss $\mathcal{L}$ from Appendix F. We fix $\theta \in \mathbb{R}^{L \times K}$ with $L = 10$ and $K = 2$. For each configuration $(t_1, n)$, we draw a batch of 500 samples $(X_i)_{i \leq 500}$ from our sampler and compute per-sample gradients $\nabla_\theta \mathcal{L}(X_i)$. Fig. 8 reports the mean and standard deviation across the batch, and matches the behavior derived theoretically: for small $t_1$, increasing $n$ substantially increases gradient dispersion, whereas for larger $t_1$ the gradient statistics become largely insensitive to $n$.

## F. Further experiments

**Polynomial programming** We illustrate our approach on the polynomial programming toy problem also considered by Tucker et al. (2017); Grathwohl et al. (2018); Paulus et al. (2020b); Liu et al. (2023a). In this setting, for all $i \in [L]$, the

distribution $\pi_\theta^i$ is given by $\pi_\theta^i = \text{Bernoulli}\left(\frac{\exp(\theta^{i1})}{\exp(\theta^{i1})+\exp(\theta^{i2})}\right)$, with $\theta \in \mathbb{R}^{L\times 2}$ (here $K = 2$). Fixing $c = 0.45$ and $p \geq 1$, we consider

$$\min_{\theta \in \mathbb{R}^{L\times 2}} \frac{1}{L} \mathbb{E}_{\pi_\theta}\left[\left\|X^{\cdot,2} - c\,\mathbf{1}_L\right\|_p^p\right], \tag{63}$$

where $X^{\cdot,2} = [X^{1,2}, \ldots, X^{L,2}]^\top$ and $\mathbf{1}_L = [1, \ldots, 1]^\top$. The minimum is attained in the limit $\theta^{i1} \to +\infty$ for all $i \in [L]$, that is when $X^{i,2} = 0$ almost surely for all $i \in [L]$. We report results in Fig. 9 with $L = 128$. We use a learning rate of $0.05$ for all methods.

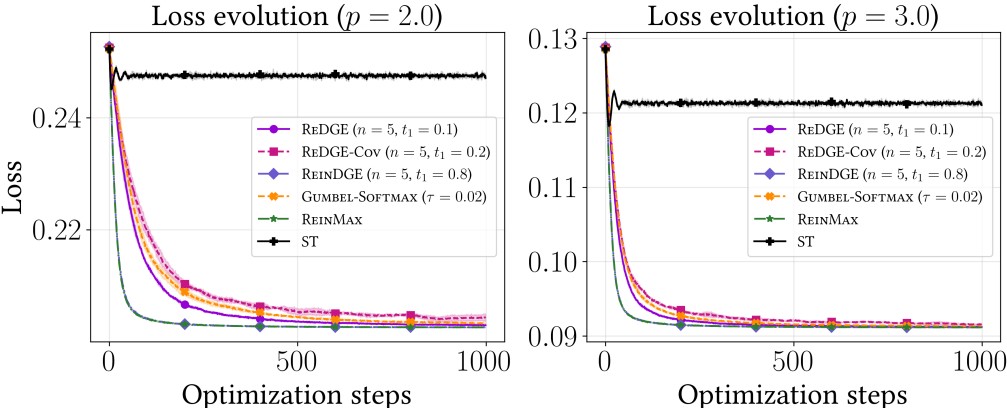

*Figure 9.* Polynomial programming benchmark for different values of the exponent $p$.

*Results.* We observe that STRAIGHT-THROUGH fails to reach the optimum and plateaus early, whereas REINMAX achieves near-optimal performance. REDGE and REDGE-COV perform on par with GUMBEL-SOFTMAX, and with appropriate hyperparameter choices converge to the correct solution (with REDGE-COV typically converging slightly faster). Finally, REINDGE also attains near-optimal performance, which is consistent with the fact that it recovers REINMAX as a special case.

Following (Liu et al., 2023a), we use a batch size of 256, a length of 128, 2 categorical dimensions and a vector $c := (c_1, \ldots, c_L) \in \mathbb{R}^L, \forall i, c_i = 0.45$.

**Remark 3.** *We highlight several limitations of this example, which, to our knowledge, have not been explicitly discussed in the gradient-estimation literature and somewhat undermine its relevance as a stand-alone evaluation:*

*(1). The objective is separable and identical across dimensions, so the gradient can be recovered from only two loss evaluations (one per coordinate value), instead of the usual $K^L$, which in this case would be $2^{128}$.*

*(2). The STRAIGHT-THROUGH estimator performs poorly in this experiment. However, note that the discrete objective is determined entirely by the values of $f$ at the vertices of the product simplex. Consequently, any extension on $\mathbb{R}^{L\times 2}$ that matches $f$ on these vertices defines the same discrete problem, yet may induce a very different optimization landscape. As an illustration, consider the extension*

$$f : x \in \mathbb{R}^{L\times 2} \mapsto \tfrac{1}{L}\sum_{i=1}^{L} |c|^p x^{i1} + |1-c|^p x^{i2},$$

*which is linear and coincides with (63) on the vertices. For this relaxation, hard ST yields a low-variance unbiased gradient estimator (and soft ST yields the exact gradient) that performs almost optimally. We show the results with this linear relaxation in Fig 10.*

*(3). Finally, REINMAX is based on a second-order Taylor approximation of $f$. Consequently, when $p = 2$ in (63), the estimator is exact (see Appendix B). For other values of $p$, the estimator is no longer exact, although it often remains a close approximation in practice. This exactness is specific to quadratic objectives and does not extend to general functions $f$.*

We see in Figure 10 that in this setting STRAIGHT-THROUGH achieves the best performance.

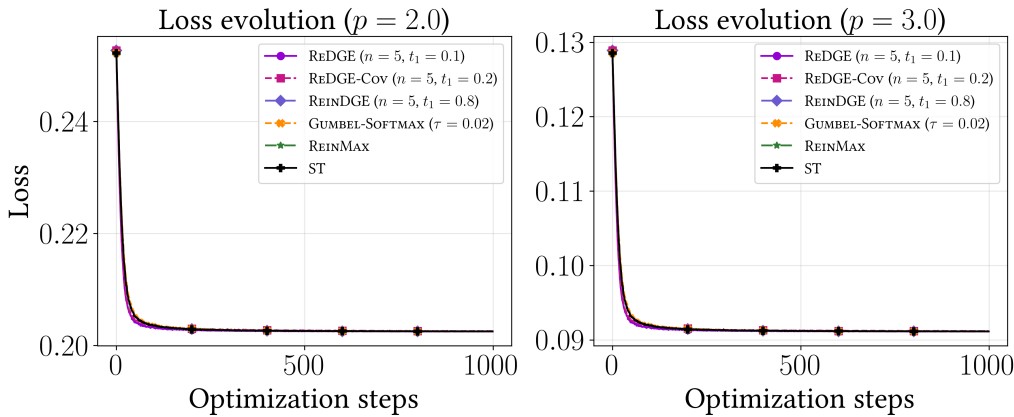

*Figure 10.* Polynomial programming benchmark for different values of the exponent $p$ with the linear relaxation.

## G. Experimental Details

### G.1. Hyperparameter sweep

For all the our diffusion samplers REDGE, REDGE-COV and REINDGE we sweep over the timestep $t_1 \in \{0.3, 0.5, 0.7, 0.9\}$, the number of diffusion steps $n - 1$ with $n \in \{3, 5, 7, 9\}$. For the GUMBEL-SOFTMAX baseline we sweep the temperature parameter $\tau \in \{0.01, 0.05, 0.1, 0.2, \dots, 1\}$. For all the baselines we also sweep over the learning rate used in Adam in the range $\{0.01, 0.05, 0.1, 0.5\}$. For all algorithms, we estimate gradients with respect to the untempered target $\pi_\theta(x) \propto \exp(\langle x, \varphi_\theta \rangle)$ and do not use the $\tau$-tempered target with p.m.f. proportional to $\exp(\langle x, \varphi_\theta \rangle / \tau)$, since this would require sweeping $\tau$ for each method and would change the objective across algorithms. Consequently, REINMAX and STRAIGHT-THROUGH require no temperature (or relaxation) hyperparameter. The hyperparameters are given in Table 5, 6 and 7.

*Table 5.* Hyperparameters with the lowest AUC of the average violations for the MDM Sudoku experiment.

| Algorithm | Hyperparameters |
| --- | --- |
| REDGE | $n = 3$, $t_1 = 0.7$, lr= 0.1 |
| REINDGE | $n = 9$, $t_1 = 0.5$, lr= 0.01 |
| REDGE-COV | $n = 5$, $t_1 = 0.9$, lr= 0.05 |
| GUMBEL-SOFTMAX | $\tau = 1.0$, lr= 0.05 |
| REINMAX | lr= 0.01 |
| ST | lr= 0.1 |

*Table 6.* Hyperparameters with the lowest AUC of the average violations for the Sudoku (without MDM) experiment.

| Algorithm | Hyperparameters |
| --- | --- |
| REDGE | $n = 3$, $t_1 = 0.5$, lr= 0.1 |
| REINDGE | $n = 9$, $t_1 = 0.5$, lr= 0.1 |
| REDGE-COV | $n = 3$, $t_1 = 0.5$, lr= 0.1 |
| GUMBEL-SOFTMAX | $\tau = 0.5$, lr= 0.1 |
| REINMAX | lr= 0.1 |
| ST | lr= 0.05 |

*Table 7.* Hyperparameters for the highest AUC for the CLIP score for the MaskGIT experiment.

| Algorithm | Hyperparameters |
| --- | --- |
| REDGE | $n = 3$, $t_1 = 0.9$, lr= 0.5 |
| REDGE-COV | $n = 3$, $t_1 = 0.9$, lr= 0.5 |
| REINDGE | $n = 3$, $t_1 = 0.9$, lr= 0.5 |
| GUMBEL-SOFTMAX | $\tau = 0.9$, lr= 0.5 |
| ST | lr= 0.5 |
| REINMAX | lr= 0.5 |

### G.2. Sudoku experiment

We follow Ye et al. (2024) and train a masked diffusion model (MDM) to approximate the distribution $p(\cdot|\mathbf{c})$ over valid completions of an incomplete Sudoku grid $\mathbf{c}$, viewed as a categorical distribution on $\mathsf{V}^{81}$; *i.e.* each cell is represented by a categorical distribution. $\mathsf{V}$ denotes the set of one-hot vectors of length 10, where the last one hot vector $e_{10}$ is reserved for the mask token $\mathtt{m}$. Let $\mathcal{G}$ be the collection of 27 groups (9 rows, 9 columns, 9 blocks), where each $g \in \mathcal{G}$ is a subset of cell indices $i \in [81]$. We define the digit-count function $s_g : X \in \mathsf{V}^{81} \mapsto \sum_{i \in \mathcal{G}} P X^i$, where $P$ removes the last coordinate of $X$ (due to the presence of the mask in the vocabulary). $s_g$ returns the all-ones vector if and only if the digits in $g$ form a valid permutation, *i.e.* each digit appears exactly once. We consider the reward $r(x) \coloneqq \sum_{g \in \mathcal{G}} \left\| s_g(x) - \mathbf{1}_9 \right\|^2$. The setup in Ye et al. (2024) consists in collect one million solved games[1] and then using the first 100k as training set and the subsequent 1000 as the testing set. The GPT-2 architecture (bi-directional) is used for the MDM with 6 million parameters. We train the model to convergence, corresponding to a solve-rate of 98%. For all methods we use a single Monte Carlo sample to estimate the loss. The hyperparameters are given in Table 5 and 6.

For the guidance algorithm, we implement Algorithm 2 using the bridge transition $q^{\mathsf{d}}_{\ell_k|0,\ell_{k+1}}(\cdot|\hat{x}_0, x_{\ell_{k+1}})$ in place of $q^{\mathsf{d}}_{\ell_k|0}(\cdot|\hat{x}_0)$. In addition, rather than sampling exactly from $\pi_\theta$ (which already performs well), we found that using its MAP estimate yields slightly better results; specifically, we set $\hat{x}_0 \leftarrow \operatorname{argmax}_{x \in \mathsf{X}} \pi_\theta(x)$. We use the schedule $\beta_{\ell_k} = 1/(k+1)$ and $M = 20$.

Finally, we make a few comments regarding the experiment without the pre-trained MDM. At first sight, taking $\pi_\theta$ to be fully factorized across cells may seem too restrictive, since valid Sudoku grids exhibit strong dependencies. The key point, however, is that while $\pi_\theta$ is mean-field *conditional on a fixed* $\theta$, the learning dynamics are not: the loss is highly non-separable, and each stochastic gradient step updates many cell logits jointly through shared row/column/block constraints. Consequently, dependencies are introduced through the optimization procedure itself. During training, updates are computed from random samples of the grid. Therefore the parameter iterate is itself random: after $T$ steps, $\theta_T$ is a random variable defined by the stochastic recursion induced by the optimizer. The distribution of an output grid produced after $T$ steps from initialization $\theta_0$ is thus the mixture $\mathbb{E}\left[\pi_{\theta_T}|\theta_0\right]$. Although each component in this mixture factorizes, the mixture does not: the shared optimization noise couples all coordinates through the non-separable constraints, allowing the resulting predictor to place most of its mass on globally consistent Sudoku configurations despite the mean-field parameterization.

### G.3. MaskGIT experiment

We study inference-time reward guidance for discrete image generation using a pretrained, class-conditional MaskGIT model (Chang et al., 2022). Concretely, we rely on the public implementation and pretrained checkpoints from Besnier et al. (2025), which operates in the discrete latent space of a VQ-VAE tokenizer. Images at resolution $384 \times 384 \times 3$ are represented as a grid of $L = 576$ discrete codes, each taking values in a codebook of size $K = 16384$. Each code is embedded in $\mathbb{R}^d$ with $d = 8$; letting $E \in \mathbb{R}^{K \times d}$ denote the embedding matrix, a token sequence $[K]^L$ corresponds to a latent embedding $x \cdot E \in \mathbb{R}^{L \times d}$ where $x \in \mathsf{X}$ is the one-hot encoding of the token sequence. The latent embedding is then decoded back to pixel space by the VQ-VAE decoder. We use a text-conditioned reward based on CLIP (Radford et al., 2021; Hessel et al., 2021): given a discrete sample $x$, we decode it to an image and compute its CLIP score with respect to a target prompt. For each baseline we performed hyperparameter sweeps over 50 images and then we ran the best configuration with 200 images; see Table 7. We use the following prompt template:

---

[1] https://www.kaggle.com/datasets/bryanpark/sudoku

**Prompt construction.** Prompts are sampled from the ImageNet class map using a deterministic template generator: We draw a class with replacement, keep the primary label (text before the first comma), and fill one of four template families—plain "a photo of a/an ..." variants, color-focused captions (45% probability), size adjectives (25%), or combined color+size (10%). Colors are restricted to CLIP-friendly adjectives (red, blue, green, yellow, black, white, silver, gold) and sizes to small, big; the remaining probability mass uses simple photographic descriptors (plain background, natural light, shallow depth of field, etc.). This yields caption-like prompts that remain grounded and avoid style cues beyond color/size.

For the guidance algorithm, we implement Algorithm 2 with two modifications: after the step $\hat{x}_0 \sim \pi_\theta$, we apply carry-over unmasking; and for the prior we use *classifier-free guidance* with guidance scale 3. Since the vocabulary $K$ is large, the covariance matrix used in the initialization of REDGE-COV becomes degenerate so we instead clamp its diagonal elements to the lowerbound 0.1.

### G.4. Runtime

We report the runtime for the guidance experiments. Our ReDGE gradient estimators trade a modest amount of extra compute for improved performance, without increasing memory.

*Table 8.* Estimated runtime for the Sudoku MDM guidance experiment on a batch of 1000 Sudokus.

| Sampler | Runtime (s) | Memory (GiB) |
|---|---|---|
| REDGE | 182.19 | 7.20 |
| REDGE-COV | 220.14 | 7.20 |
| REINDGE | 199.12 | 7.20 |
| ST | 154.35 | 7.20 |
| GUMBEL-SOFTMAX | 150.00 | 7.20 |
| REINMAX | 160.00 | 7.20 |

*Table 9.* Estimated runtime the MaskGIT guidance experiment on a batch of 10 images.

| Sampler | Runtime (s) | Memory (GiB) |
|---|---|---|
| REDGE | 1118.78 | 8.23 |
| REDGE-COV | 1168.27 | 8.23 |
| REINDGE | 1163.91 | 8.23 |
| ST | 1128.24 | 8.23 |
| GUMBEL-SOFTMAX | 1095.44 | 8.23 |
| REINMAX | 1105.54 | 8.23 |

```python
def reindge(logits_theta, T_t1_theta, n_steps, eta, **kwargs):
    # categorical denoiser
    denoiser_fn = partial(cat_denoiser, logits=logits_theta)

    schedule_kwargs = kwargs["schedule"]

    # base noise X_1 ~ N(0, I) used by the (approx.) transport map
    X1 = torch.randn(*logits_theta.shape, requires_grad=True, device=logits_theta.
    device)

    # transport to time t1, then last-step categorical objects
    # returns: hard sample X0, relaxed proxy \hat{X0} (for ST), and proxy used in
    forward pass where theta is detached
    X0_hard, X0_soft_hat, X0_theta_detached = T_t1_theta(
        logits=logits_theta,
        initial_noise=X1,
        denoiser_fn=denoiser_fn,
        n_steps=n_steps,
        eta=eta,
        **schedule_kwargs,
    )

    delta = (X0_hard - X0_soft_hat).detach()

    grad_proxy_storage = {"value": None}
    ran_once = {"done": False}
    handles = {}

    def save_grad_proxy(grad):
        # stores upstream gradient dL/d(X0_theta_detached)
        grad_proxy_storage["value"] = grad.detach()
        return grad

    def add_reinmax_to_logits(grad_logits):
        grad_proxy = grad_proxy_storage["value"]

        if not ran_once["done"]:
            ran_once["done"] = True

            # ST part: gradient through the relaxed surrogate \hat{X0}
            g_ST_logits = torch.autograd.grad(
                outputs=X0_soft_hat,
                inputs=logits_theta,
                grad_outputs=grad_proxy,
                retain_graph=True,
            )[0]

            # projection term along delta = X0 - \hat{X0}
            inner = (grad_proxy * delta).sum(dim=-1, keepdim=True)
            proj_delta = inner * delta

            # ReinMax-style correction (as in Eq. (grad-reinmax) in the paper)
            g_RM_logits = 0.5 * (g_ST_logits + proj_delta.squeeze())

            handles["h_proxy"].remove()
            handles["h_logits"].remove()
            return grad_logits + g_RM_logits

        return grad_logits

    handles["h_proxy"] = X0_proxy.register_hook(save_grad_proxy)
    handles["h_logits"] = logits_theta.register_hook(add_redgemax_to_logits)

    return X0_theta_detached
```

*Figure 11.* Implementation of REINDGE(§B.3) via PyTorch hooks.

