# OpenReview forum: "Categorical Reparameterization with Denoising Diffusion Models"
_ICML.cc/2026/Conference — ICML 2026 regular_

### Official Review · Reviewer_QwsV · 2026-03-03

**Soundness:** 2
**Presentation:** 3
**Significance:** 2
**Originality:** 3
**Overall Recommendation:** 5
**Confidence:** 3

**Summary:**

This paper proposes a novel gradient estimator for optimization problems whose objectives are expectations over categorical distributions. Intuitively, we would like to use the reparametrization trick, and represent the categorical distribution as the pushforward of a simpler base distribution through some learned map. However, the naive approach fails for discrete distributions, which motivates continuous relaxations like the Gumbel softmax trick. This paper proposes a gradient estimator called ReDGE based on an alternative continuous relaxation that uses closed-form DDIM iterations to sample from an approximation to the discrete distribution. The authors propose some extensions of their estimator and demonstrate its efficacy on a number of benchmarks drawn from generative modeling and Sudoku solving.

**Compliance With Llm Reviewing Policy:**

Affirmed.

**Final Justification:**

This is a well-written paper presenting a reasonable gradient estimation strategy for optimization problems whose objectives are expectations over categorical distributions. The authors complement good empirical results with a reasonable analysis of their method's theoretical properties. As the authors addressed my concerns re: the necessity of DDIM sampling in their rebuttal, I have raised my score for this paper.

**Key Questions For Authors:**

I'm not sure if I understand why DDIM sampling is necessary to implement the ReDGE estimator, and I would appreciate if the authors could help me clarify my understanding. In particular, it seems to me that the proposed strategy is equivalent to applying Gaussian smoothing to $\pi_theta$ and then applying the reparametrization trick. By assumption, $\pi_\theta$ is discrete and there is some map $T_\theta(z)$ such that $T_\theta(z) \sim \pi_\theta$. The DDIM sampling map at time $t_k$, which the authors call $T^\theta_{t_k}$, has the property that $T^\theta_{t_k}(z)$ has the same distribution as the RV $\alpha_t X_0 + \sigma_t X_1$. In other words, the following sampling procedures are equivalent in distribution:

1. Sample Gaussian noise $z$, run DDIM sampling, output $T^\theta_{t_k}(z))$.
2. Sample Gaussian noise $x$, compute $T_\theta(z)$, output $\alpha_t T_\theta(z) + \sigma_t \epsilon$, where $\epsilon$ is standard normal.

In particular, expectations over these two distributions are equivalent, and since the gradient (6) we're interested in is expressed as an expectation over either of these distributions, we can approximate it using samples from the second procedure, which should be much cheaper than DDIM sampling. I might be wrong here and would appreciate clarifications.

One potential typo: In the statement of the ReDGE estimator, should $T^\theta_0(X_1)$ read $T^\theta_{t_1}(X_1)$? Isn't the point of Proposition 1 that we *don't* want $t_1 = 0$?

**Limitations:**

Yes

**Strengths And Weaknesses:**

This is a well-written paper, and I was able to follow the main ideas without too much trouble. The authors adequately situate their work in the context of the literature on gradient estimation for learning models with categorical variables. The theoretical results in this paper (essentially Proposition 1) are correct to the best of my knowledge, and the approach itself is reasonable. The results reported for the experiments are also strong. I am persuaded that this method is a reasonable strategy for gradient estimation with categorical variables. However, I have concerns about the necessity of performing DDIM sampling to implement the continuous relaxation of $\pi_\theta$, which I detail below. I am prepared to raise my score for this paper if the authors can clarify my understanding and address my concerns about the proposed method.

---

> ### Author Rebuttal · Authors · 2026-03-31
>
> We thank the reviewer for the careful and constructive review. We are glad that you found the paper well-written overall and that you found the empirical results strong. We address the remaining concern regarding the DDIM sampling below.
>
> We agree with that the two procedures are equivalent **in distribution**. However, the proposed construction relies on an exact sampler $T_\theta$ for the categorical distribution. As discussed in the paper, such exact reparameterization are piecewise constant in $\theta$, so their Jacobian vanishes almost everywhere and the reparameterization formula does not apply in the discrete setting.
>
> So the key issue is not how to sample the Gaussian-smoothed marginal, but how to construct a smooth, efficient, and differentiable approximate reparameterization from a Gaussian base distribution to the categorical distribution. For instance, if we were to use the procedure you described, we wouldn't be able to differentiate the samples with respect to theta. We use DDIM because it provides a simple, training-free, differentiable transport built entirely from Gaussian operations, and in our setting the denoiser is available in closed form. Moreover, this construction comes with a natural relaxation parameter that directly controls the sharpness of the transport, and thus the trade-off between fidelity to the categorical distribution and gradient degeneracy. Finally, as we argued to Reviewer MoPr, this reparameterization is less prone to numerical precision errors compared to Gumbel-Softmax.
>
> If the reviewer had a specific example of such a reparameterization in mind, we would be very grateful to hear it, as this would help us tailor our response more precisely.
>
> Regarding the second point concerning the definition of the ReDGE estimator, our use of $T^\theta _0(X_1)$ is not a typo; by definition we have that $T^\theta _0(X_1) = T^\theta _{0|t_1}(T^\theta _{t_1}(X_1))$. So our point is not that we should avoid the map to time 0 altogether, we simply need to ensure that $t_1$ is not close to $0$ so that the Jacobian of the map $T^\theta _{0|t_1}$ is well-behaved and doesn’t vanish.
>
> Again, we thank the reviewer for the thoughtful and constructive feedback, and we remain open to any further questions or suggestions that may help us improve the paper.

---

> > ### Author Rebuttal · Reviewer_QwsV · 2026-04-03
> >
> > Thank you for clearing up these points of confusion. I appreciate your clarifications and have raised my score accordingly.

---

> > > ### Author Response · Authors · 2026-04-07
> > >
> > > Thank you for the follow-up and for your thoughtful reconsideration. We appreciate your openness to our clarifications and are glad that they helped resolve the points of confusion.

---

### Official Review · Reviewer_MoPr · 2026-03-09

**Soundness:** 3
**Presentation:** 2
**Significance:** 3
**Originality:** 4
**Overall Recommendation:** 3
**Confidence:** 2

**Summary:**

This paper proposes a new method for deriving a path-wise reparameterization gradient for optimizing through sampling operations, inspired by diffusion models. They claim that their method matches or outperforms existing methods.

**Compliance With Llm Reviewing Policy:**

Affirmed.

**Key Questions For Authors:**

- Overall, I don't understand the choices in L212-216 in the right column. Why does $T_{0|t_1}$ takes $X_1$ as input? $1 \neq t_1$ so their time steps don't match.
- How do you sample from $\pi_{0|t_1}$ conditioned on $T_{0|t_1}(X_1)$? DDIM is deterministic, so I'm not sure how you get multiple samples from the same starting point $T_{0|t_1}(X_1)$. Do you use any other latent variable than $X_0$ and $X_1$ to sample from this distribution (e.g., reverse diffusion process)?

**Limitations:**

I don't see the limitation section in the current manuscript.

**Strengths And Weaknesses:**

Strength

- The idea of using a closed-form optimal diffusion model for deriving the reparameterization gradient is novel.
- They claim that the proposed method can interpolate between known methods, such as the straight-through estimator, which can give a unified picture.


Weakness

- As noted by the authors, the idea of getting a reparameterization gradient with a biased distribution already exists: Gumbel softmax. I don't see why this method should outperform Gumbel softmax.
- Experiment results are mixed, and oftentimes the proposed method does not show a clear benefit over others (the difference is small when it exists). The authors argue that their method's performance is more consistent, but this may be due to the fact that the proposed method has more hyperparameters to tune (t1 and n), and that for any dataset, there always are some hyperparameters that work,  which can be unfair.

---

> ### Author Rebuttal · Authors · 2026-03-31
>
> We thank the reviewer for the constructive feedback and for recognizing the novelty of the method. We address the main concerns below.
>
> >Why ReDGE can outperform Gumbel-Softmax.
>
>  Beyond the empirical results, we outline three reasons.
> 1. ReDGE is more stable in low precision.
> Gumbel-Softmax relies on the transform $-\log(-\log U)$ followed by a temperature-scaled softmax. When $\tau$ is small, perturbations in the noise or logits are amplified by the $1/\tau$ factor, so in low precision (e.g. BF16) rounding errors can lead to noticeably different gradients.
> To illustrate this, we ran an additional diagnostic on Sudoku guidance. For each sampler/configuration, we followed its own FP32 optimization trajectory for 1000 optimization steps on the 1000-puzzle validation batch. At each step, we fixed the current logits, drew 20 noise samples, and computed gradients in both FP32 and BF16. We then measured gradient cosine, relative gradient error, BF16 cosine-to-mean, and BF16 coordinate standard deviation:
>
> | sampler | grad cosine | rel grad err | BF16 cos-to-mean | BF16 coord std |
> |---|---:|---:|---:|---:|
> | redge (n = 3, t_1=0.5, lr=0.1) | 0.9931 | 0.1166 | 0.8281 | 0.0510 |
> | gumbel\_tau=1.0\_lr=0.1 | 0.9822 | 0.1882 | 0.3911 | 0.1975 |
> | gumbel\_tau=1.0\_lr=0.3 | 0.9716 | 0.2357 | 0.4251 | 0.1425 |
> | gumbel\_tau=0.5\_lr=0.3 | 0.9492 | 0.3134 | 0.3961 | 0.1998 |
>
> These results suggest that ReDGE has both smaller BF16-vs-FP32 gradient drift and substantially lower gradient variance along the actual optimization trajectory.
>
> Furthermore, in low precision the Gumbel-max trick may no longer accurately sample from $\pi_\theta$ because of numerical error. This issue with Gumbel-based sampling is known in the literature; see [1, Sec. 5.2–5.3]. To illustrate this, we compared the empirical law produced by ReDGE ($t_1=0.5$, $n=5$) and Gumbel samplers to $\pi_\theta$:
>
> | Sampler | KL mean | KL std | L2 mean | L2 std |
> |---|---:|---:|---:|---:|
> | gumbel_float64  | 0.000233 | 0.000100 | 0.006097 | 0.001946 |
> | redge_float64   | 0.000235 | 0.000089 | 0.006366 | 0.001587 |
> | gumbel_bfloat16 | 0.000843 | 0.000668 | 0.009162 | 0.002618 |
> | redge_bfloat16  | 0.000239 | 0.000109 | 0.006681 | 0.002153 |
>
> 2. The main ReDGE relaxation parameter is easier to tune and less sensitive. The Gumbel temperature $\tau$ is not easily interpretable and takes values in the literature ranging from very small (0.01) to quite large ($\geq 5$), often with unpredictable sensitivity. By contrast, the analogous ReDGE parameter $t_1$ always lies on the interpretable scale $0,1$. In our experiments, a coarse sweep (0.2) was already sufficient, and values around $0.5$ worked well as a default. As we show in the ImageNet experiment, strong ReDGE performance persists across $t_1 \in {0.5, 0.7, 0.9}$, whereas Gumbel-Softmax is substantially more sensitive to temperature.
>
> 3. ReDGE recovers useful edge cases. ReDGE has a practical advantage: as $t_1$ approaches 1, it moves toward ST, which is itself a strong baseline. Gumbel-Softmax does not enjoy the same behavior: increasing $\tau$ may stabilize gradients, but when $\tau$ becomes too large the relaxation approaches noise rather than a useful surrogate.
>
> Hyperparameter concern
>
> We agree that ReDGE has one additional method parameter compared with Gumbel-Softmax, so this is a fair point to raise. However, we do not think the comparison is unfair in practice, because the two parameters do not play equally important roles. The parameter $n$ is not a strong free parameter in practice. The paper already shows that increasing the number of diffusion steps has little effect except in the very sharp regime of very small $t_1$, and Appendix E.2 studies this interaction in more detail. In other words, ReDGE behaves much more like a method with one main tuning parameter plus a weak secondary discretization parameter. Consistent with this, fixing $n=3$ already gives good performance across our experiments, so one can treat it as fixed unless additional compute is available for tuning; see the heatmap in Figure 4.
>
> >Questions
>
> Thank you for pointing out the typo. It should be $T^\theta _{t_1}(X_1)$;  i.e. an approximate sample $X _{t_1}$ from $\pi^\theta _{t_1}$  as mentioned in line 208 of the right column. As for your second question, we sample $X_0$ from the transition $\pi^\theta _{0|t_1}$ as we have access to it in closed form; it is a factorized categorical distribution with probability vector given by $\hat{x}^\theta _0(\cdot, t_1)$. This step is only used to have a hard sample that is correlated with the soft sample with respect to which we differentiate ($T^\theta _{0}(X_1))$. However, the different samples come from running the ddim procedure from different initial noises.
>
> We thank the reviewer for their time and constructive review and remain open to questions or suggestions.
>
>  [1] Zheng et al., Masked Diffusion Models Are Secretly Time-Agnostic Masked Models and Exploit Inaccurate Categorical Sampling.

---

> > ### Author Rebuttal · Reviewer_MoPr · 2026-04-03
> >
> > Concerns partially resolved.
> >
> > 1. I still think performance improvement is not large enough.
> > 2. I think the numerical precision argument regarding Gumbel softmax is rather weak. One may use various tricks, such as enabling fp64 only for the softmax layer with negligible cost, which is already mentioned [1].
> >
> > [1] Zheng et al., Masked Diffusion Models Are Secretly Time-Agnostic Masked Models and Exploit Inaccurate Categorical Sampling.

---

> > > ### Author Response · Authors · 2026-04-07
> > >
> > > We thank the reviewer for the follow-up and for engaging further with our rebuttal. We believe we have addressed the main concerns, including hyperparameters and comparison to Gumbel-Softmax. While we understand the reviewer’s reservations we believe that the paper already provides enough evidence to support our claims.
> > >
> > > Empirically, ReDGE appears to offer a meaningful advantage. Across the paper’s benchmarks and the added rebuttal experiments, ReDGE consistently matches or outperforms Gumbel-Softmax, sometimes by a large margin. In Sudoku, ReDGE reaches nearly 80% accuracy and ReDGE-Cov 100%, versus about 70% for Gumbel-Softmax. In large-scale ImageNet, ReDGE also achieves a clearly higher CLIP score. Given the breadth of comparisons and careful baseline tuning, we believe the results show a clear practical improvement.
> > >
> > > At the same time, our contribution is not only empirical. ReDGE reflects a genuinely different view of gradient estimation for categorical distributions, yielding a novel methodology rather than a minor variant of existing approaches. Its significance lies both in the performance gains and in introducing a new methodological framework that we further analyse theoretically.
> > >
> > > Overall, we believe the paper offers both a novel, technically grounded estimator and clear empirical improvements over the most relevant baseline and we hope that this can be reflected in the final assessment.

---

### Official Review · Reviewer_WAZL · 2026-03-10

**Soundness:** 2
**Presentation:** 3
**Significance:** 3
**Originality:** 3
**Overall Recommendation:** 5
**Confidence:** 4

**Summary:**

This paper considers the optimization problem $\min_{\theta} E_{\pi_{\theta}}(f(X))$, which typically arises from the reinforcement learning context. Here $f$ is given and $\pi_{\theta}$ is a categorical distribution and is assumed to be factorizable over $X = (X^1,X^2,\ldots,X^L)$. In order to optimize the above objective, we usually require the computation of the gradient w.r.t. $\theta$, $\nabla_{\theta} E_{\pi_{\theta}}(f(X))$. This gradient is given by $\sum_{x \in \mathcal{X}}{f_{\theta}(x) \nabla_{\theta} \pi_{\theta}(x)}$ where $\mathcal{X}$ is the state space of the size $K^L$ (L variables, each has $K$ states) -- which is intractable. Traditional methods like REINFORCE construct stochastic approximation for this sum, but have limitations. In this work, the key contribution is as follows: the authors propose a new way to approximately compute this gradient based on diffusion models. The starting idea is to use \emph{reparameterization trick} (Kingma & Welling, 2013): **if** $\pi_{\theta}$ can be written as $\pi_{\theta} = Law(T_{\theta}(Z))$ for some pushfoward map and $Z$ is a simple distribution (Gaussian), the objective function can be written as $E_{Z}(f(T_{\theta}(Z)))$ and -- under some regularity -- the gradient operator can be swap with the expectation, resulting in $\nabla_{\theta} E_{Z}(f(T_{\theta}(Z))) = E_Z(J_{\theta} T_{\theta}(Z)^{\top} \nabla_x f(T_{\theta}(Z)))$. Then, the key idea of this paper is to use a diffusion model to construct $T_{\theta}$ (because they are of the same nature, transforming noise to data). In particular, in this setting, the denoiser (of the diffusion model) has a closed-form solution. The newly proposed gradient estimator is called ReDGE.

**Compliance With Llm Reviewing Policy:**

Affirmed.

**Final Justification:**

The authors sufficiently addressed my main concerns. I believe this is a solid paper with rigorous mathematical grounding. The considered problem is important and fundamental for the ML community. I gave 4 (weak accept) mainly because I believe a more complete analysis of the proposed estimator within a framework of a canonical optimization scheme (e.g., SGD) is truly important for understanding its advantage.

**Update after the last rebuttal by the authors**

I thank the authors for making a great effort in addressing my comment. I encourage the authors to integrate this new result into the paper, possibly as a remark, as I believe it is important in its own right. I therefore increase my score from 4 to 5.

**Key Questions For Authors:**

- Does the proposed $T_{\theta}$ satisfy the domination condition so you can swap gradient and expectation?
- How much different is your closed-form derivation of the denoiser from those in the context of Dirac deltas?
- Do you have any insight on the variance of the proposed estimator? Since it is in the reparametrization regime, it automatically has good variance, but does diffusion model provide further improvement?
- How about the code? Do you have it anywhere?

**Limitations:**

Yes, to some extent. It would be better to have a separate paragraph for it.

**Strengths And Weaknesses:**

# Strength
- The idea is elegant: using a diffusion model to approximate the reparameterization map is appealing in several senses. There are no obvious reparametrization maps we can construct by hand in this setting. Even if there is one, it will be ill-posed (vanishing gradient) since the target distribution is categorical. The diffusion model allows some smoothness relaxation: we can control how close to the target distribution by indicating the number of diffusion steps.
- The method recovers STRAIGHT-THROUGH when using 1 diffusion step.
- The denoiser has a closed-form and simple solution, so no need to train the diffusion model at all.
- The method works well in a number of experiments, and they are more stable with comparable or better performance than other baselines.

# Weakness
- The factorization of the target distribution limits its area of application.
- As a relaxation method, it still violates the categorical constraints, although the violation rate is low.
- The overall analysis appears incomplete: it is unclear how to quantify the overall optimization algorithm (e.g., SGD) using this ReDGE gradient estimator.

---

> ### Author Rebuttal · Authors · 2026-03-31
>
> We thank the reviewer for the positive assessment. We are glad the reviewer found the idea elegant and the experiments strong.
>
> >On the factorization assumption.
>
> We agree that, probabilistically, a fully factorized (mean-field) target cannot represent inter-dimensional correlations in one step. But this is not specific to our method. It is standard and essentially necessary in categorical reparameterization: Gumbel-Softmax (Jang et al., 2017; Maddison et al., 2017) and related variants (Tucker et al., 2017) use the same setup. Computationally, relaxing the joint law of L variables with K states requires a simplex of dimension $K^L-1$, so both softmax and normalization scale exponentially. The tractable alternative is therefore to factorize into L independent $(K-1)$-dimensional simplexes. This need not limit applicability, since expressivity depends on state-space design. If local dependencies must be modeled explicitly, one can interpolate between full independence and a full joint model via block factorizations. Grouping pairs gives $L' = 2$ blocks with $K' = K^{L/2}$ states each. Larger blocks increase expressivity at higher computational cost. Appendix G.2 gives further intuition for why factorization can still be effective for Sudoku despite its strong dependencies.
>
> >On violation of categorical constraints.
>
> Our method does satisfy variable-level categorical constraints. As stated in Section 4, the hard variant of REDGE yields a one-hot matrix in the forward pass, so the objective only receives valid discrete states. If the reviewer instead refers to domain-level violations (e.g. Sudoku rules), we agree: our method is not a deterministic constraint solver. However, the low violation rate indicates that the relaxation still provides accurate optimization guidance for difficult discrete problems.
>
> >Analysis of the complete algorithm.
>
> A full SGD-level analysis of REDGE would be valuable, but it would require tracking how estimator bias and variance propagate through the iterates, which is beyond the scope of this paper. We therefore focus on the aspect most specific to REDGE and most relevant in practice: how the relaxation parameter $t_1$ affects gradient informativeness. We view this as a first step toward a fuller optimization theory.
>
> >Domination condition
>
> Our estimator satisfies this condition under standard regularity assumptions on $f$. One can control the Jacobian of $T^\theta _0$ following the ideas in the proof of Proposition 1 in Appendix A, where an explicit bound is obtained. The key point is that the DDIM update has the form $x_s=a(s,t) \hat{x}^\theta _0(x_t, t) + b(s,t) x_t$, with $\hat{x}^\theta _0(x_t, t)=softmax(\theta+c(t)x_t)$. Since $t_1>0$, the derivatives of the denoiser with respect to both $\theta$ and $x_t$ are uniformly bounded, and a simple recursion over the finitely many DDIM steps yields a uniform bound on the Jacobian of $T^\theta _0(z)$ on any neighborhood $U$ of $\theta$, say $\sup _{\theta \in U} | \partial _\theta T^\theta _{0}(z)| \le C_U$. Moreover, because $\hat{x}^\theta _0(x_t, t)$ lies in the simplex, its norm is uniformly bounded, and the same recursion gives $|T^\theta _{0} (z)|\le A+B|z|$ for some constants $A,B>0$. By the mean-value theorem, for any direction $h$, $| \frac{f(T^{\theta+h} _{0}(z))-f(T^\theta _{0} (z))}{h} | \le C _U \sup _{\theta \in U} | \nabla f(T^\theta _{0}(z)) |$. Assuming that $f \in C^1$ and that $| \nabla f(x) | \le C(1+|x|^m)$ for all $x\in\mathbb R^L$, we obtain $|\frac{f(T^{\theta+h} _0(z))-f(T^{\theta} _0(z))}{h} | \le C_U C (1+(A+B|z|)^m )$. We integrate this bound over a standard Gaussian; since it has finite moments of every order, the right-hand side is integrable. This yields an $L^1$-dominating function and justifies interchanging expectation and gradient by dominated convergence.
>
> >Closed-form denoiser
>
> If the target is a mixture of Dirac masses on embeddings, ie $\pi_\theta=\sum_{i=1}^K w_\theta^i,\delta_{X_0^i}$ with $X_0^i \in \mathbb R^d$, then the denoiser is the posterior mean $\mathbb E[X_0\mid X_t=x_t]=\sum_{i=1}^K \hat{x}^\theta _0 (x_t, t)^i X_0^i$, so each reverse step requires posterior weights over the K atoms (to compute $\hat{x}^\theta _0 (x_t, t)$) and a codebook multiplication. In contrast, our method uses a diffusion process on the one-hot labels $J \sim \sum^K _{i=1} w^i _\theta \delta _{e_i}$ so the denoiser is simply $\hat{x}^\theta _0(x_t, t)$, without an additional matrix multiplication at each step. Embeddings, if needed, are recovered only at the end by matrix multiplication.
>
> >Variance
>
> We have run an additional low-precision experiment comparing the variance to Gumbel-Softmax to illustrate the benefits of our method. Please see our reply to Reviewer MoPr.
>
> >Code
>
> The code for all algorithms and experiments in the paper is here: (https://anonymous.4open.science/r/temp_redge-4D3A).
>
> We thank the reviewer for their time and constructive review and remain open to questions or suggestions.

---

> > ### Author Rebuttal · Reviewer_WAZL · 2026-04-02
> >
> > Thank you for taking the time to address my remarks. I maintain a positive assessment of the manuscript; however, I believe it would be more complete with the inclusion of an analysis of the optimization scheme.

---

> > > ### Author Response · Authors · 2026-04-08
> > >
> > > We thank the reviewer for this comment and appreciate that they continue to view the paper positively. We would like to emphasize that a full theoretical analysis of the optimization scheme is well beyond what is usually included in works of this kind. Such a study typically requires a different viewpoint and substantially different tools from those used to analyze the estimator itself, and often constitutes a research direction in its own right. In particular, this level of analysis is not provided in the original Gumbel-Softmax paper [1], nor, to the best of our knowledge, in closely related works on gradient estimators for discrete variables [2,3,4,5]. When such results do appear, they are usually the main focus of separate follow-up works, or are obtained only in highly simplified settings (for example $L=1, K=2$); see, e.g., [6]. In our case, however, the detailed study of the gradient carried out in Appendix A already provides the key ingredients needed to derive this type of result, and we outline below how such an analysis can be obtained.
> > >
> > > As we cannot provide the proof in a PDF as per the ICML rules, we lay below the core argument of the proof. Concretely, we study the ReDGE estimator $g_{t_1}(\theta;X_1)=J_{\theta} T^{\theta} _0(X_1)^T \nabla_x f(T_0^{\theta}(X_1))$, where $X_1$ is standard Gaussian and $t_1$ is the final diffusion time under the idealized setting where there is no discretization error in the interval $[t_1, 1]$. This setting is already challenging and the full analysis including the discretization error warrants a paper on its own. In order to get an SGD bound, we need to bound the bias and the expected squared norm of our gradient estimator.
> > >
> > > First, under the no discretization assumption, the expectation of our gradient is equal to
> > > $$\mathbb{E}[g_{t_1}(\theta;X_1)] = \mathbb{E} _{\pi^{\theta} _{t_1}} [\nabla _{\theta}  \log \pi^{\theta} _{t_1} (X _{t_1}) f(\hat{x}^\theta _0(X _{t_1}, t_1)) + J _\theta \hat{x}^\theta _0(X _{t_1}, t_1) ^\top \nabla _x f(\hat{x}^\theta _0(X _{t_1}, t_1))]
> > > $$
> > > We have used a similar decomposition to derive the ReinDGE estimator in section 3.3. On the other hand, we can write the true gradient as $\mathbb{E} _{\pi^\theta _0}[\nabla _\theta \log \pi^\theta _0(X_0) f(X_0)]$. The difference of both gradients can be studied using two crucial arguments; first: the target score identity $\nabla _\theta \log \pi^\theta _{t _1}(x _{t _1}) = \mathbb{E}[\nabla _\theta \log \pi^\theta _0(X _0) | X _{t _1} = x _{t_1}]$ from [7] which allows us to bound the norm
> > > $\\mathbb{E}[\| \nabla _\theta \log \pi^\theta _{t _1} (X _{t _1}) - \nabla _\theta \log \pi^\theta _0 (X _0) \|]$  and also the norm of the denoiser Jacobian $\| J _\theta \hat{x}^\theta _0(., t _1)\|$. Second, these quantities can be bounded by $\mathbb{E}[1 - \pi^\theta _{0|t _1}(X_0 | X _{t _1})]$ which itself is bounded by $C \exp(-\alpha _{t _1} \Delta^2/ \sigma^2 _{t _1})$ where $\Delta$ is a constant. Under common smoothness assumptions on $f$ we then get an exponentially vanishing bias in $t_1$.
> > >
> > > On the other hand, to bound the expected norm of the gradient we extend Proposition 2 from the Appendix of our paper to the no discretization setting to get an upper-bound on the expected squared norm of the gradient that grows as $\alpha _{t _1} / \sigma^{2} _{t _1}$, under the same smoothness assumptions on $f$. By combining these results with the descent lemma for smooth functions, we get the **biased SGD bound** on a uniform iterate $R$ in $[1:N]$; with a specific step-size depending on $t_1$ and $N$:
> > > $$
> > >     \mathbb{E} \| \nabla F(\theta _R)\|^2 \lesssim \sqrt{(1 + \alpha _{t _1} / \sigma^2 _{t _1}) / N} + \exp(-2 C \alpha _{t _1} / \sigma^2 _{t _1})
> > > $$
> > >
> > > This bound makes the temperature trade-off explicit. As $t_1$ decreases, the bias decays exponentially fast, while the stochastic term grows through the effective inverse temperature $\alpha _{t _1}/\sigma _{t _1}^2$. Thus, lower temperature improves approximation to the discrete objective at the cost of higher optimization noise.
> > >
> > > We believe this is the result the reviewer was expecting. Thank you again for pointing this out.
> > >
> > > [1] Jang et al., Categorical Reparameterization with Gumbel-Softmax.
> > > [2] Fan et al., Training Discrete Deep Generative Models via Gapped Straight-Through Estimator.
> > > [3] Grathwohl et al., Backpropagation through the Void.
> > > [4] Liu et al., Bridging Discrete and Backpropagation: Straight-Through and Beyond.
> > > [5] Bengio et al., Estimating or Propagating Gradients Through Stochastic Neurons for Conditional Computation.
> > > [6] Shekhovtsov, Bias-Variance Tradeoffs in Single-Sample Binary Gradient Estimators .
> > > [7] De Bortoli, V., Hutchinson, M., Wirnsberger, P. and Doucet, A., 2024. Target score matching.

---

### Official Review · Reviewer_NudS · 2026-03-10

**Soundness:** 3
**Presentation:** 4
**Significance:** 3
**Originality:** 3
**Overall Recommendation:** 4
**Confidence:** 4

**Summary:**

This paper presents REDGE, a diffusion-based soft reparameterization method for categorical distributions. The approach derives practical gradient estimators, including hard variants that recover **STRAIGHT-THROUGH** and **REINMAX** as one-step special cases. The paper also provides theoretical analysis on how the diffusion parameter $t_1$ controls the relaxation level and affects gradient behavior. Then they evaluate it on several benchmarks, including tasks in discrete optimization and models with categorical latent variables.

**Compliance With Llm Reviewing Policy:**

Affirmed.

**Final Justification:**

I am maintaining my score at 4. The rebuttal clarified my questions about the relaxation parameter and practical tuning, and I still find the paper technically solid and novel. My main remaining concern is that the scalability evidence is still somewhat limited: the ImageNet experiment is useful, but it is only one large-scale setting. Overall, I think this is a good paper with clear merit, but this keeps me at weak accept rather than a higher score.

**Key Questions For Authors:**

1. The diffusion parameter $t_1$ plays an important role in controlling the relaxation level and gradient behavior. How sensitive is the method to different choices of $t_1$, and is there practical guidance for selecting it in different settings?

2. How does the method scale when the categorical dimension becomes large (e.g., large vocabularies or large action spaces)?

**Limitations:**

Yes.

**Strengths And Weaknesses:**

Strength:
1. Clear problem formulation and motivation: The paper studies the problem of categorical reparameterization and proposes REDGE to obtain pathwise gradient estimators. The motivation and formulation are clearly presented.
2. Solid theoretical analysis: The theoretical analyses explains well why sharper relaxations can lead to uninformative gradients.

Weakness:
1. Sensitivity to the diffusion parameter $t_1$:  The diffusion parameter $t_1$ plays an important role in controlling the relaxation level and gradient behavior. However, the paper provides limited empirical analysis on how sensitive the method is to different choices of $t_1$.

2. Scalability to large categorical spaces: The proposed method performs diffusion on the simplex whose dimension grows with the number of categories.The paper does not discuss how the method scales when the categorical dimension becomes large.

---

> ### Author Rebuttal · Authors · 2026-03-31
>
> We thank the reviewer for the thoughtful and positive assessment, and we are glad that the problem formulation, theoretical analysis, and overall empirical results came across clearly. We address the remaining concerns below.
>
> > On the relaxation parameter $t_1$
>
> We agree this is an important question, and we would like to clarify that the paper already addresses it in several places. Proposition 1 theoretically formalizes why taking $t_1$ too small makes gradients uninformative, which we also confirm experimentally. In particular, Fig. 4 reports a full heatmap over $(n ,t_1)$, Fig. 5 plots ImageNet CLIP performance as a function of $t_1$ and compares it to Gumbel-Softmax temperature sensitivity, and Appendix E.2 further studies the interaction between $t_1$ and $n$. We acknowledge that this may not have been sufficiently highlighted in the main paper and we will revise the paper accordingly. Taken together, these results support a fairly clear practical message: performance is stable for moderate $t_1$, while very small $t_1$ is consistently harmful. This is also why we summarize a simple tuning guideline in the last paragraph before the conclusion: use a small number of diffusion steps ($n=3$ or $5$) together with a moderate $t_1$ (typically $0.5$, $0.7$, or $0.9$). In the revision, we can make this practical guidance more explicit in the main text. We hope this clarifies our practical guideline for $t_1$, and we would be happy to further emphasize any aspect of the sensitivity analysis that the reviewer feels is still missing.
>
> > Scalability to large categorical spaces
>
> We would also like to emphasize that the paper addresses this particular issue raised by the reviewer to a meaningful extent: it discusses the numerical challenges that arise for large $K$, which motivates the scalar-variance variant for ReDGE-Cov, and it includes an ImageNet experiment in a large categorical regime (L=576, K=16384). Empirically, our experiments cover a fair range of dimensions ( L=128, K=2 for polynomial programming, L=81, K=9 for sudoku and L=576, K=16384 for ImageNet). We hope that this clarifies that our method is indeed tested in a fairly large scale setting.
>
> Finally, our experiments suggest a simple practical picture for choosing the hyperamater $t_1$ when we scale the dimension; first the appropriate choice of $t_1$ does not seem to depend strongly on the number of variables $L$, since $\pi_\theta$ factorizes across coordinates, but it becomes more important to avoid overly small $t_1$ as the vocabulary size $K$ grows. Intuitively, when $t_1$ is too small, the proxy gradient that we consider only reflects one or a few coordinates of the categorical distribution, which reduces its usefulness and increases variance. Using a larger $t_1$ retains information about a broader set of class probabilities and leads to more stable gradients.
>
> We again thank the reviewer for their time and constructive review, and we remain open to any further questions or suggestions.

---

> > ### Author Rebuttal · Reviewer_NudS · 2026-04-03
> >
> > Thank you for the rebuttal. My concerns have been adequately addressed, especially the discussion around the choice of the diffusion parameter $t$ and the applicability of the method in larger discrete settings. I appreciate the clarifications and maintain my positive assessment of the paper. I therefore maintain my current score, while encouraging the authors to make the practical scalability discussion a bit more explicit in the main text.

---

> > > ### Author Response · Authors · 2026-04-07
> > >
> > > Thank you for the thoughtful follow-up and for your positive assessment of our work. We are glad that our rebuttal has provided clarification. We also appreciate your suggestion regarding scalability, and we will make this discussion more explicit in the main text.

---

### Decision · Program_Chairs · 2026-04-30

**Decision:**

Accept (regular)

**Comment:**

This paper introduces a novel method to obtain a gradient estimator, for a discrete distribution using a continuous relaxation that works well in a number of experiments. There are some concerns around the motivation and whether certain baselines could be computed in higher numerical precision. The novelty, empirical results and otherwise clarity of writing outweigh those concerns.